# Knothe-Rosenblatt Quantile Regression for Risk-sensitive Multi-objective Reinforcement Learning

**Gwangpyo Yoo** [1]   **Woo Kyung Kim** [1]   **Honguk Woo** [1]

## Abstract

In this work, we extend distributional reinforcement learning (RL) to develop a risk-sensitive multi-objective RL framework, with applications to domains such as finance and robotics. We achieve this by adopting vector-risk measures and approximating them via Knothe-Rosenblatt (KR) quantile regression. This approach directly extends the IQN framework to the multi-objective setting, aligns with the axiomatic definition of vector-risk measures, and guarantees that critics converge under the distributional Bellman operator. To mitigate the artificial ordering imposed by the KR map, we employ a transformer architecture without positional encoding, and introduce MO-TQC for training stability. We demonstrate improved performance on MO-Gymnasium benchmarks and use our framework to study risk-sensitive policies in multi-objective tasks.

## 1. Introduction

Reinforcement learning (RL) has proven effective in various sequential decision-making problems, demonstrating significant success in domains such as finance (Zhang et al., 2020; Wang & Ku, 2022), autonomous driving (Bernhard et al., 2019), robotics (Bodnar et al., 2020), and healthcare (Lu et al., 2020). However, these real-world applications often require highly sophisticated reward engineering, which remains a substantial barrier to the practical deployment of RL. Furthermore, these domains are inherently risk-sensitive. For instance, a robotic agent is expected to complete tasks while minimizing energy consumption, whereas a financial trading agent must maximize profits while mitigating trading costs or potential losses. These conflicting objectives require risk-sensitive decision-making, as failures in these domains often lead to asymmetric or irreversible consequences.

The most intuitive way to overcome such compound problems is to apply multi-objective RL (MORL) and risk-sensitive RL simultaneously. However, within the machine learning community, there has been little theoretical or practical discussion regarding risk-sensitive MORL, with the exception of distributional MORL methods that utilize the duality of stochastic dominance in DPMORL (Cai et al., 2023). To achieve multi-objective risk-sensitive RL, four key components are required:

1. The definition of multivariate risk.
2. A framework that tractably handles risk.
3. Convergence guarantees for the critic in RL.
4. Practical engineering to facilitate better convergence.

Fortunately, the definition of multivariate risk has already been discussed in the field of financial mathematics (Ararat & Feinstein, 2024; Jouini et al., 2004; Hamel et al., 2007); vector-risk (Jouini et al., 2004). The vector-risk measure serves as a natural extension of univariate risk measure concepts to the multivariate setting. This function computes an expectation vector that prioritizes adverse outcomes by emphasizing worse case scenarios for each objective in an elementwise manner[1]. Here, a risk measure refers to the function used to calculate risk.

In risk-sensitive single-objective RL, calculating risk via a weighted integral of the quantile function has become a prominent approach (Dabney et al., 2018a). Similarly, in MORL, we aim to compute vector-risk using a weighted integral of the quantile function. A significant challenge, however, is that the definition of a quantile for a random vector is ambiguous. We adopt the Knothe-Rosenblatt (KR) map as a multivariate quantile formulation. Since it has been unknown whether this calculation aligns with the theoretical definition of vector-risk, we provide a proof of this correspondence and demonstrate the convergence of using the KR map as a critic in RL. Note that this is distinct from prior MMD-based methods (Zhang et al., 2021; Wiltzer et al., 2024) or marginal-based strategies (Kim et al., 2023; Zhou et al., 2021). MMD-based methods are not order-preserving

---

[1]Department of Computer Science and Engineering, Sungkyunkwan University, Suwon, South Korea. Correspondence to: Honguk Woo <hwoo@skku.edu>.

*Proceedings of the 43rd International Conference on Machine Learning*, Seoul, South Korea. PMLR 306, 2026. Copyright 2026 by the author(s).

---

[1]Although this is technically a coherent risk; an inaccurate description is provided here for better understanding.

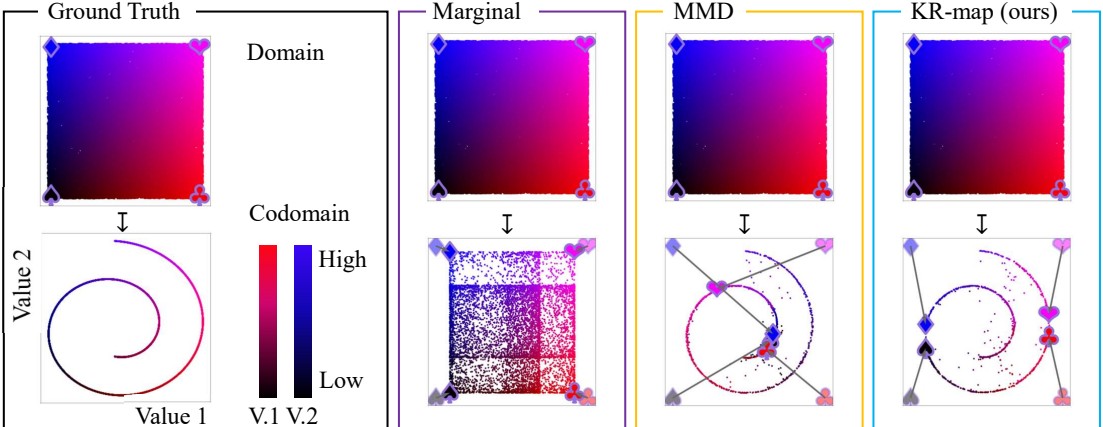

*Figure 1.* **Comparison of marginal, MMD, and KR map approaches.** The upper area is the domain of the particle (quantile-level for KR map) and the lower is the codomain of the map, each of which aims to regress ground truth (left side). The gray arrows describe how the vertices of the domain, as a box, are mapped. As an MMD-based approach is not a transport map, the arrows are crossing; calculating risk is not appropriate.

(i.e., not a transport map), rendering them unsuitable for computing risk. Further, they have a theoretical flaw (Dabney et al., 2018b). See Corollary B.1 in Appendix B. Meanwhile, marginal-based approaches fail to capture correlations among objectives and suffer from convergence issues. See Figure 1 for details.

In essence, the KR map is an autoregressive quantile function. Therefore, it inherently imposes an order among objectives. This induced ordering can be unnatural in scenarios where objectives lack a predefined hierarchy, thereby imposing an incorrect inductive bias on the model. As a result, the model may capture spurious correlations among objectives, reducing performance. To address this issue, we use a transformer-based architecture without positional encoding (PE). Its autoregressive property enables flexible modeling of the conditional distributions in the KR map. By integrating this architecture with our theoretical findings, we propose a framework, KR Implicit Quantile Network (KR-IQN). This framework enables the agent to learn dependency structures and perform risk-sensitive decision-making in multi-objective environments.

Finally, to ensure practical convergence and stability, we propose Multi-objective TQC (MO-TQC). Building on the success of TQC (Kuznetsov et al., 2020a) in mitigating overestimation bias and controlling distributional precision via ensemble-based backpropagation, MO-TQC extends this truncation mechanism to the multivariate setting, reducing preference-induced overestimation that can remain after scalarization. Our contributions are as follows:

- We propose KR-IQN, the first practical framework that enables tractable risk-sensitive decision-making in MORL settings.

- We establish a theoretical foundation for risk-sensitive MORL by proving that the KR map aligns with the rigorous definition of vector-risk and demonstrating the convergence of the KR-IQN.

- To mitigate the artificial ordering of the KR map, we propose a transformer-based architecture without PE for KR map estimation.

- We introduce MO-TQC, which incorporates dimension-wise and summation atom removal mechanisms, to address overestimation bias and ensure stability in multi-objective settings.

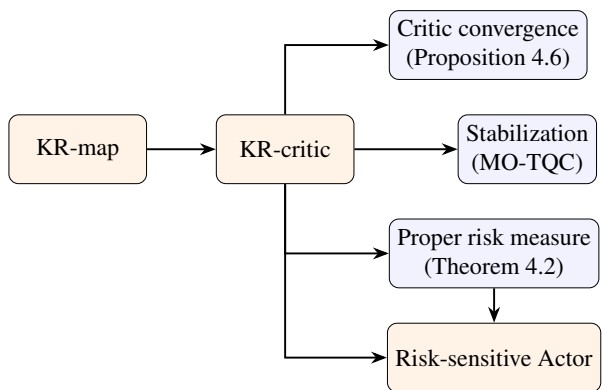

*Figure 2.* Structure of major contributions

Section 2 reviews related work. Section 4 proves that the risk measure induced by the KR-map is a proper risk measure and states the convergence of the KR-critic-based Bellman equation. Section 5 details the practical implementation and introduces multi-objective TQC to stabilize the learning process. Section 6 presents the experimental results in

MO-Gymnasium along with a case study in a financial application. An overview of how these proposed components depend on one another is visually summarized in Figure 2.

## 2. Related Work

**MORL:** Existing distributional MORL methods face limitations in risk assessment. MMD-based approaches (Zhang et al., 2021; Wiltzer et al., 2024) lack the transport property essential for risk calculation and sample minimizer problem (see Appendix B), while marginal-based frameworks (Zhou et al., 2021; Kim et al., 2023) fail to capture objective dependencies. Utility-based methods like DPMORL (Cai et al., 2023) suffer from scalability issues (Röpke et al., 2023). Our work builds on the non-distributional CAPQL (Lu et al., 2023), utilizing its preference-conditioned policy with entropy maximization, in contrast to policy family approaches like PGMORL (Xu et al., 2020) or DPMORL.

**Risk-sensitive RL:** In single-objective risk-sensitive RL, the framework corresponding to ours is the IQN framework proposed by Dabney et al. (2018a). We have extended what Dabney et al. (2018a) accomplished by performing quantile regression in one dimension and computing distortion risk measures through KR quantile regression. Regarding practical implementation, Yoo & Woo (2025); Yoo et al. (2024) previously utilized TQC in risk-sensitive RL. Notably, Yoo & Woo (2025) provided an analysis of the positive effects of TQC in risk-sensitive offline RL.

**Multivariate Quantile and Risk:** Various multivariate quantiles exist (Abdous & Theodorescu, 1992; Kong & Mizera, 2012; Carlier et al., 2016; Wei, 2008), but most (e.g., spatial, directional) are set-valued, resulting in difficulty for deep RL. While the optimal transport (OT, Carlier et al. (2016)) offers vector-valued map with the transport property, it has computational and theoretical limitations for RL. We thus adopt the KR map (Wei, 2008) to facilitate tractable risk calculation. Regarding vector-risk, we follow the established vector-based framework (Hamel et al., 2007; Ararat & Feinstein, 2024). Note that this is within the existing set-valued framework, with the cone fixed as $\mathbb{R}_0^+$. It induces an elementwise order and simplifies the intractable set-valued map for a deep RL agent into a tractable vector-valued map.

## 3. Preliminary and Problem Formulation

### 3.1. Preliminary

We consider spectral risk measures. The spectral risk measure is a class of risk measures that emphasize worse cases. Therefore, spectral risk measures are defined as weighted integrals of the quantile function of the return distribution.

**Definition 3.1.** *Let $X$ be a random variable with cumulative distribution function (CDF) $F_X : \mathbb{R} \to [0, 1]$. The quantile function of $X$, denoted by $F_X^{-1} : [0, 1] \to \mathbb{R}$, is defined as the left inverse of $F_X$; i.e.,*

$$F_X^{-1}(u) = \inf\{x \in \mathbb{R} : F_X(x) \geq u\}, \quad u \in [0, 1]. \quad (1)$$

**Definition 3.2.** *(Acerbi, 2002) Let $\mathcal{Z}$ be a set of random variables denoting returns. Let $\phi : [0, 1] \to [0, \infty]$ be a non-increasing weight function satisfying $\int_0^1 \phi(p)\, dp = 1$ (i.e., $\phi$ is some probability density). The spectral risk measure $\rho(\cdot; \phi) : \mathcal{Z} \to \mathbb{R}$ is defined as*

$$\rho(Z; \phi) = -\mathbb{E}_{u \sim U[0,1]}[F_Z^{-1}(\Phi^{-1}(u))], \quad (2)$$

*where $\Phi^{-1}$ is the quantile function of $\phi$; i.e., $F_\phi^{-1}$.*

For example, Conditional Value at Risk (CVaR) with parameter $\alpha \in (0, 1)$ is defined with $\phi(u; \alpha) = \frac{1}{\alpha}\mathbf{1}_{u \leq \alpha}(u) \Leftrightarrow \Phi^{-1}(u; \alpha) = \alpha u$.

We introduce the quantile loss for distributional critics.

**Theorem 3.3.** *(Univariate Quantile Loss; Koenker (2005)) Let $Z$ be a random variable whose mean is finite. Fix $u \in (0, 1)$. Let $\mathcal{L}_{QR}$ be a loss function defined as*

$$\mathcal{L}_{QR}(\theta; Z) = \mathbb{E}[|u - \mathbb{I}(Z - \theta < 0)| \times |Z - \theta|]. \quad (3)$$

*The problem $\min_\theta \mathcal{L}_{QR}(\theta; Z)$ is a convex minimization problem. The solution exists and is equal to the quantile $F_Z^{-1}(u)$.*

**Definition 3.4.** *(Knothe-Rosenblatt map[2], Knothe (1957); Rosenblatt (1952)) Let $\mathbf{Z}$ be a $d$-dimensional random vector. For readability, we denote the conditional quantile as $F_X^{-1}(u; y)$ rather than $F_{X|Y=y}^{-1}(u)$. A map $\mathbf{Q_Z} : [0, 1]^d \to \mathbb{R}^d$ is called the Knothe–Rosenblatt map (KR map) if it is defined as*

$$\begin{aligned}
&[\mathbf{Q_Z}(\mathbf{u})]_1 = F_{\mathbf{Z}_1}^{-1}(\mathbf{u}_1), \\
&[\mathbf{Q_Z}(\mathbf{u})]_k = F_{\mathbf{Z}_k}^{-1}(\mathbf{u}_k; [\mathbf{Q_Z}(\mathbf{u})]_{<k}) \text{ for } k = 2, \ldots, d.
\end{aligned} \quad (4)$$

*Here $\mathbf{u} \in [0, 1]^d$, and $\mathbf{X}_{<k}$ denotes $[\mathbf{X}_1, \ldots, \mathbf{X}_{k-1}]$.*

**Theorem 3.5.** *The KR map $\mathbf{Q_Z}(\mathbf{u}) \sim \mathbf{Z}$ if $\mathbf{u} \sim U[0, 1]^d$.*

This is known as the pushforward property. For the proof, see (Backhoff et al., 2017).

Considering the conditional quantile and univariate case, we can consider the quantile loss for a KR map, KR quantile regression, by minimizing the loss as follows.

**Corollary 3.6.** *(KR Quantile Loss) Let $\mathbf{u} \in (0, 1)^d$, and $\mathbf{Z}$ be a $d$-dimensional random vector, whose first moment is finite. We define KR-quantile loss $\mathcal{L}_{KR}$ as*

$$\mathcal{L}_{KR}(\boldsymbol{\theta}; \mathbf{Z}, \mathbf{u}) = \sum_{k=1}^{d} \mathbb{E}[|\mathbf{u}_k - \mathbb{I}([\mathbf{Z}_k|\mathbf{Z}_{<k}] - \boldsymbol{\theta}_k < 0)| \quad (5)$$
$$\times |[\mathbf{Z}_k|\mathbf{Z}_{<k}] - \boldsymbol{\theta}_k|],$$

---

[2]It is also known as KR rearrangement or KR quantile.

with the convention that $\mathbf{Z}_1|\mathbf{Z}_{<1} := \mathbf{Z}_1$. *From the Theorem 3.3 and Theorem 3.5, we observe that the solution of* $\min_{\boldsymbol{\theta} \in \mathbb{R}^d} \mathcal{L}_{KR}(\boldsymbol{\theta}; \mathbf{Z})$ *exists and is equal to* $\mathbf{Q}_\mathbf{Z}(\mathbf{u})$.

## 3.2. Problem Formulation

We consider a multi-objective MDP (MO-MDP). MO-MDP is a 5-tuple $(\mathcal{S}, \mathcal{A}, \mathbf{R}, \mathcal{P}, \gamma)$, where $\mathcal{S} \neq \emptyset$ is a state space, $\mathcal{A} \neq \emptyset$ is an action space, $\mathbf{R} : \mathcal{S} \times \mathcal{A} \to \mathbb{R}^d$ denotes rewards, which may be random, $d$ is the number of objectives, $\mathcal{P} : \mathcal{S} \times \mathcal{A} \to \mathbb{P}(\mathcal{S})$ is a transition kernel, and $\gamma \in [0, 1)$ is a discount factor. Here, $\mathbb{P}(X)$ denotes the set of all probability measures defined on the set $X$.

Let $\pi : \mathcal{S} \to \mathbb{P}(\mathcal{A})$ be a policy. Define multi-objective state-action value distribution $\mathbf{Z}^\pi(s_t, a_t)$ as

$$\mathbf{Z}^\pi(s_t, a_t) := \sum_{i=0}^{\infty} \gamma^i \mathbf{R}(s_{t+i}, a_{t+i}) \tag{6}$$

where $s_t$ is a $t$-timestep state and $a_t \sim \pi(s_t)$.

Assume that a preference set $\mathcal{W} \subseteq \triangle$ is given, where $\triangle := \{\sum_{i=1}^d \omega_i = 1, \omega_i \geq 0 \text{ for any } i = 1, \ldots, d\}$. Further, a set of univariate risk measures $\{\rho(\cdot; \phi_i)\}_{i=1}^d$, associated with each objective, is also given. Define $\boldsymbol{\Phi}^{-1} : [0, 1]^d \to [0, 1]^d$ as

$$\boldsymbol{\Phi}^{-1}(\mathbf{u}) = [\Phi_1^{-1}(\mathbf{u}_1), \ldots, \Phi_d^{-1}(\mathbf{u}_d)], \tag{7}$$

where $\Phi_i^{-1}$ is the quantile function of $\phi_i$. Let $\mathbf{Q}_{\mathbf{Z}^\pi} : [0, 1]^d \to \mathbb{R}^d$ be the KR map for $\mathbf{Z}^\pi$. Our objective is to find a policy $\pi_\omega^*(s, \cdot)$ for each preference,

$$\pi_\omega^*(s_t, a_t) = \arg\min_\pi -\omega^T \mathbb{E}_{\mathbf{u} \sim U[0,1]^d}[\mathbf{Q}_{\mathbf{Z}^\pi} \circ \boldsymbol{\Phi}^{-1}(\mathbf{u})] \tag{8}$$

for any $\omega \in \mathcal{W}$. This is a natural extension of the risk-sensitive single-objective RL to the MORL.

# 4. Theoretical Analysis

Here, we prove that Eq.(8) is a proper vector-risk measure and that a critic learned from KR-Quantile regression converges. We also propose exotic risk measures, which do not exist in a univariate space.

## 4.1. Vector-risk Measure

**Definition 4.1.** *(dynamic vector-risk measure, (Jouini et al., 2004)) Let $\leq_e$ be an elementwise inequality defined on $\mathbb{R}^d$. Let $L_d^\infty(\mathcal{F}_T)$ be a set of random variables that we are interested in. Here $\mathcal{F}_T$ is a filtration (history) of the coordinate process. A map $\rho_t : L_d^\infty(\mathcal{F}_T) \to L_d^{(t)}$ is called a (dynamic) vector-risk measure if it satisfies:*

(Z) $\rho_t(\mathbf{0}) = \mathbf{0}$,
(M) $\mathbf{X} \leq_e \mathbf{Y}$ a.s. $\Rightarrow \rho_t(\mathbf{Y}) \leq_e \rho_t(\mathbf{X})$,
(C) $\rho_t(\mathbf{X} + \mathbf{c}) = \rho_t(\mathbf{X}) - \mathbf{c}$ for any constant $\mathbf{c} \in \mathbb{R}^d$.

**Theorem 4.2.** *Define $\rho_\mathbf{\Phi}(\mathbf{X}) := -\mathbb{E}_{\mathbf{u} \sim U[0,1]^d}[\mathbf{Q}_\mathbf{X} \circ \boldsymbol{\Phi}^{-1}(\mathbf{u})]$. Then, there exists a dynamic risk measure $\{\rho_k\}_{k=1}^d$ such that $\rho_\mathbf{\Phi} = \rho_d$.*

**Proof**: (Z) and (C) are trivial. For (C), $\mathbf{Q}_{\mathbf{X}+\mathbf{c}}(\mathbf{u}) = \mathbf{Q}_\mathbf{X}(\mathbf{u}) + \mathbf{c}$. The negation follows from the Eq.(8).

To prove (M), we need to introduce an extended concept. Note that this does not change any framework or theoretical perspective in our work. We introduce this concept because, although the KR-map is effective for modeling a single random vector, it struggles in comparative settings involving multiple vectors. Such settings necessitate comparisons of unrealistic pairs[3], or require excessively restrictive assumptions like comonotonicity or identical copulas.

**Definition 4.3.** *(Causal Transport (Beiglböck et al., 2024; Acciaio et al., 2020)) Consider a random vector's elements as a realization of the random process (coordinate process) $\mathbf{X} = (\mathbf{X}_1, \ldots, \mathbf{X}_d)$. The causal transport is defined as*

$$[\mathbf{Q}_\mathbf{X}(\mathbf{u})]_k = \mathbf{Q}_\mathbf{X}(\mathbf{u}_k|\mathcal{F}_{<k}), \tag{9}$$

*where $\mathcal{F}_{<k}$ denotes an arbitrary history (filtration) that can be obtained before the $k$-th step. Note that if we choose $\mathcal{F}_k = \mathbf{X}_{<k}$, then this coincides with the KR map.*

Choose $\mathcal{F}_{<k} = (\mathbf{X}_{<k}, \mathbf{Y}_{<k})$. Then $(\mathbf{Q}_\mathbf{X}(\mathbf{u}), \mathbf{Q}_\mathbf{Y}(\mathbf{u}))$ follows Law$(\mathbf{X}, \mathbf{Y})$ by the pushforward property of the regular KR map. Further, $\mathbf{Q}_\mathbf{X}(\mathbf{u}_k) \leq_e \mathbf{Q}_\mathbf{Y}(\mathbf{u}_k)$. Since the image of $\boldsymbol{\Phi}^{-1}$ is contained in $[0, 1]^d$, the property (M) holds.

The actor loss defined in Eq.(8) does not introduce a comparison of risk between two policies, nor does the critic loss in Eq.(5). Therefore, Def. 4.3 is purely conceptual[4]

## 4.2. Extended Risk Measures

As an MORL framework, we may extend the $\boldsymbol{\Phi}$ in Eq.(7), which induces a risk measure. Consider a map $\boldsymbol{\Phi}^{-1}_{\alpha\triangle} : [0, 1]^d \to \alpha\bar{\triangle}^d$, where $\alpha\bar{\triangle} := \{\alpha\mathbf{x}| \|\mathbf{x}\|_1 \leq 1\}$, $\alpha \in (0, 1]$ is constant; e.g., $\boldsymbol{\Phi}^{-1}_{\alpha\triangle}(\mathbf{x}) = \alpha\frac{\|\mathbf{x}\|_\infty}{\|\mathbf{x}\|_1}\mathbf{x}$.

**Definition 4.4.** *The simplex risk measure is defined as $\rho_\triangle(\mathbf{X}; \alpha) := -\mathbb{E}_{\mathbf{u} \sim U[0,1]^d}[\mathbf{Q}_\mathbf{X} \circ \boldsymbol{\Phi}^{-1}_{\alpha\triangle}(\mathbf{u})]$, where $\alpha \in [0, 1]$.*

**Corollary 4.5.** *(Corollary of Theorem 4.2) The simplex risk measure $\rho_\triangle(\cdot; \alpha)$ is a proper vector-risk measure.*

The simplex risk measure is one example of a non-trivial extension of the one-dimensional CVaR risk measure to the multivariate case. Note that when $\alpha = 1$, it considers the case where one of the returns can be optimistically high, but not others simultaneously.

---

[3]Pairs $\mathbf{X}, \mathbf{Y}$ with zero probability of joint realization.

[4]However, one should implement the causal transport to incorporate the risk-comparison with KR map.

## 4.3. Convergence of KR-Quantile Regression

Define distributional MO-Bellman operator $\mathcal{T}^\pi(\cdot)$ as following (Wiltzer et al., 2024; Zhang et al., 2021),

$$\mathcal{T}^\pi \mathbf{Z}^\pi(s_t, a_t) = \mathbf{R}(s_t, a_t) + \gamma \mathbf{Z}^\pi(s_{t+1}, a_{t+1}). \quad (10)$$

Since the Wasserstein distance does not depend on field properties or order properties of the real numbers, but on metric properties, we can enjoy many results of distributional RL theory proposed by (Bellemare et al., 2017; Dabney et al., 2018b; Zhou et al., 2023) and others, though we must replace the standard Wasserstein with the adapted Wasserstein (Wasserstein distance conditioned on the coordinate process) to accommodate the KR-map.

In particular, the distributional Bellman operator directly induces a valid multi-objective temporal difference (TD) target, allowing us to apply KR quantile-regression based updates without additional assumptions. Given this TD-target, we show that the critic converges by minimizing the KR quantile loss in Eq.(5).

**Proposition 4.6.** *Let* $\mathbf{Z}, \mathbf{Z}' \in \mathcal{Z}$ *be two state-action value distributions in an MO-MDP with countable state and action spaces. Then, for any policy* $\pi$,

$$\bar{d}_\infty(\Pi_{\mathrm{AW}} \mathcal{T}^\pi \mathbf{Z}, \ \Pi_{\mathrm{AW}} \mathcal{T}^\pi \mathbf{Z}') \ \le \ \gamma \bar{d}_\infty(\mathbf{Z}, \mathbf{Z}'), \quad (11)$$

*where*

$$\bar{d}_\infty(\mathbf{Z}, \mathbf{Z}') = \sup_{s,a \in \mathcal{S} \times \mathcal{A}} AW_\infty\big(\mathbf{Z}(s,a), \mathbf{Z}'(s,a)\big). \quad (12)$$

*In other words,* $\Pi_{\mathrm{AW}} \mathcal{T}^\pi$ *is a* $\gamma$-contraction on $(\mathcal{Z}, \bar{d}_\infty)$. *The definitions of* $\Pi_{\mathrm{AW}}$ *and* $AW_\infty$ *are in Appendix A.1.*

The adapted Wasserstein distance $AW$ is a variant of the Wasserstein distance for an autoregressive structure and $\Pi_{\mathrm{AW}}$ is the quantile projection operator for the empirical KR-map. The proof follows the contraction argument of Dabney et al. (Dabney et al., 2018b), tailored to the multi-objective setting. See Appendix A for details.

## 5. Implementation of KR-IQN

Our approach is directly inherited from IQN and CAPQL. First, a preference vector $\omega \in \triangle$ is sampled from the Dirichlet distribution $\mathrm{Dirch}(1, 1, \ldots, 1)$, so that a uniform distribution of preferences can be learned. This approach does not require any a priori assumption about the preference, therefore it is general enough to apply to any MO-MDP. For simplicity, we denote the $\theta$-parameterized critic as $\mathbf{Q}_{\mathbf{Z}^\pi}(\mathbf{u}; s_t, a_t, \omega, \theta)$. Here, the preference vector $\omega$ is given as an input of the critics, enabling the critics to incorporate the actor's behavioral changes into the value function. Then, for the critic learning, we calculate the TD-target with parameter $\theta^{\mathrm{target}}$ by

$$\mathbf{Q}^{\mathrm{target}} = \mathbf{R}(s_t, a_t) + \gamma \mathbf{Q}_{\mathbf{Z}^\pi}(\mathbf{u}; s_{t+1}, a_{t+1}, \omega, \theta^{\mathrm{target}}), \quad (13)$$

following distributional MO-Bellman Equation in Eq.(10). For simplicity, we omit the entropy maximization terms used in our actual implementation.

The critics are trained to minimize KR quantile loss between $\mathbf{Q}_{\mathbf{Z}^\pi}(\mathbf{u}; s_t, a_t, \omega, \theta)$ and $\mathbf{Q}_{\mathrm{TQC}}^{\mathrm{target}}$ as defined in Eq.(5)

$$\theta = \mathrm{argmin}_\theta \mathbb{E}_{\mathbf{u} \sim U[0,1]^d} \mathcal{L}_{\mathrm{KR}}(\theta; \mathbf{Q}_{\mathrm{TQC}}^{\mathrm{target}}, \mathbf{u}), \quad (14)$$

so that Eq.(10) can be achieved. The details about TQC are explained in Section 5.2. Finally, the actor can achieve the objective Eq.(8) by the loss function,

$$\mathcal{L}(\pi) = -\omega^T \mathbb{E}_{\mathbf{u} \sim U[0,1]^d}[\mathbf{Q}_{\mathbf{Z}^\pi}(\mathbf{\Phi}^{-1}(\mathbf{u}); s, a, \omega, \theta)]. \quad (15)$$

Since we do not introduce a target actor, we identify $\pi$ as a policy parameter. Algorithm 1 depicts the overall procedure.

---

**Algorithm 1** KR-IQN Learning

---

**Rollout stage:**
$\pi$ : Policy, $\mathcal{D}$ : Replay Buffer, env : environment.
$\omega \leftarrow \mathrm{Dirich}(1, \ldots, 1)$
$s_t \leftarrow \mathrm{reset}(\mathrm{env}), \mathrm{done} \leftarrow \mathrm{False}$
**while** not done **do**
    $a_t \leftarrow \pi(s_t)$
    $s_{t+1}, \mathbf{R}(s_t, a_t), \mathrm{done} \leftarrow \mathrm{step}(\mathrm{env}, a_t)$
    $\mathcal{D}.add(s_t, a_t, \mathbf{R}_t, s_{t+1})$
**end while**
**Critic Learning:**
$\theta$: param critic, $\theta^{\mathrm{target}}$: param critic target, $\lambda_c$: critic lr
$\tau$: soft update coefficient
$\omega \leftarrow \mathrm{Sample}(\mathrm{Dirich}(1, \ldots, 1))$
$\mathbf{u} \leftarrow \mathrm{Sample}(U[0, 1]^d)$
$s_t, a_t, \mathbf{R}(s_t, a_t), s_{t+1} \leftarrow \mathrm{sample}(\mathcal{D})$
$a_{t+1} \leftarrow \mathrm{Sample}(\pi(s_{t+1}))$
$\mathbf{Q}^{\mathrm{target}} \leftarrow \mathbf{R}(s_t, a_t) + \gamma \mathbf{Q}_{\mathbf{Z}^\pi}(\mathbf{u}; s_{t+1}, a_{t+1}, \omega, \theta^{\mathrm{target}})$
$\mathbf{Q}_{\mathrm{TQC}}^{\mathrm{target}} \leftarrow \mathrm{TQC}(\mathbf{Q}^{\mathrm{target}})$.
/* **KR Quantile Regression by Eq.(5)***/
$\theta \leftarrow \theta - \lambda_c \nabla_\theta \mathcal{L}_{\mathrm{KR}}(\mathbf{Q}_{\mathbf{Z}^\pi}(\mathbf{u}; s_t, a_t, \omega, \theta); \mathbf{Q}_{TQC}^{\mathrm{target}}, \mathbf{u})$
$\theta^{\mathrm{target}} \leftarrow (1 - \tau)\theta^{\mathrm{target}} + \tau\theta$
**Actor Learning:**
$\mathbf{\Phi}^{-1}$ : Risk Measure, $\lambda_a$: actor lr
$a \leftarrow \mathrm{Sample}(\pi(s, \omega)), \mathbf{u} \leftarrow \mathrm{Sample}(U[0, 1]^d)$
$\mathcal{L}(\pi) \leftarrow -\omega^T \mathbb{E}_{\mathbf{u} \sim U[0,1]^d}[\mathbf{Q}_{\mathbf{Z}^\pi}(\mathbf{\Phi}^{-1}(\mathbf{u}); s, a, \omega, \theta)]$.
$\pi \leftarrow \pi - \lambda_a \mathcal{L}(\pi)$

---

### 5.1. KR Quantile Regression with Transformers

Although the KR map guarantees a pushforward map, it generates an artificial ordering relationship. To relax the problem, we adopt a transformer for the value embedding without PE, and shuffle the objective order when training the critic. Figure 3 depicts the overall architecture.

The backbone's input consists of state, action, preference, and the value inferred for previously estimated objectives.

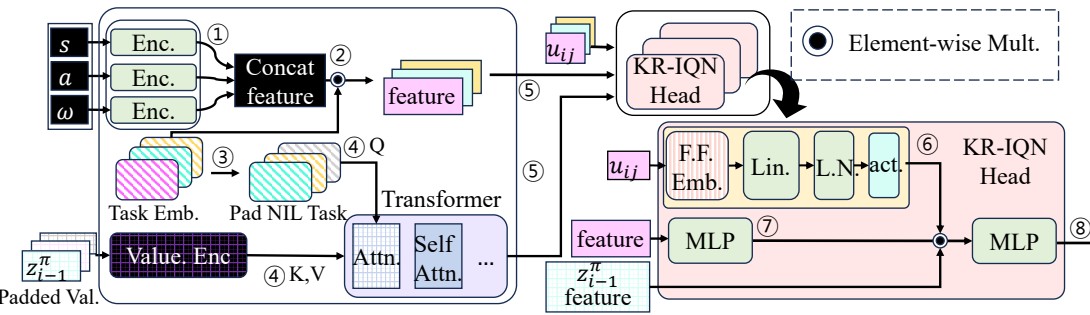

*Figure 3.* Overall architecture of our proposed method. F.F. emb denotes the Fourier Feature, Lin. denotes the Linear layer, and L.N. denotes the Layer normalization.

Internally, there are task embeddings for denoting what kind of objectives are being estimated. We use the terminology task embedding rather than objective embedding, as the latter may be mistaken for the objective values themselves or prior predictions. This separation of value predictions and task embeddings ensures distinction between padding and objectives, and among objectives with value estimates of similar scales.

First, the state, action, and preference are input through individual encoders, and their features are concatenated. Then, they are repeated and elementwise multiplied to match the currently inferred objective. This enables the critic to understand which objective is currently being inferred. Next, task embeddings are shifted and padded with NIL task. Then, they are fed to the transformer with encoded previous value predictions. In this way, the inferred values for each previous dimension required for the KR rearrangement can be used auto-regressively as input. PE is not used for the transformer to minimize the disadvantage of KR rearrangement, which enforces an unnatural ordering. Finally, the value is estimated via the KR-IQN head, which is essentially an IQN head receiving an additional feature: the previous value prediction.

### 5.2. Multi-objective TQC

As demonstrated in (Kuznetsov et al., 2020b; Yoo & Woo, 2025), TQC benefits outlier removal and distribution precision control by back-propagating over ensembles rather than a single model. We propose MO-TQC, which introduces two truncation steps: marginal TQC (dimension-wise atom removal) and projective TQC (summation atom removal). The intuitive approach is projective TQC, applying truncation to the projected distribution $\omega^T \mathbf{Q}^{\text{target}}$. However, overestimated quantile atoms associated with low-weighted objectives may persist and induce overall action-value overestimation. Moreover, since objectives in MORL are often negatively correlated, simply enforcing underestimation may be undesirable. Thus, MO-TQC first truncates upper quantiles (and optionally lower quantiles) in each marginal

distribution, then truncates the projected quantiles.

The first step is to find the largest and smallest TD-target atoms for each objective and remove them. The next step is to compute the linearization $\omega^T \mathbf{Q}^{\text{target}}$ for a given preference $\omega$ and remove the largest atom.

## 6. Experiment

We compare with the baselines introduced below using MO-Gymnasium (Felten et al., 2023), which is the standard benchmark for MORL. We also evaluate KR-IQN in financial trading environments using FinRL (Liu et al., 2020). All results are averaged over 5 seeds. See Appendix D for the hyperparameter details and environment settings.

**Baselines: PGMORL** (Xu et al., 2020) is a PPO-based algorithm that first learns a preference-conditioned policy during a warm-up phase, and then evolves specialized preference-specific policies during the evolutionary phase. **DPMORL** (Cai et al., 2023) is a PPO-based algorithm that aims to obtain a diverse set of optimal policies with respect to model-generated and non-decreasing utilities, seeking to achieve stochastic dominance across them. **CAPQL** (Lu et al., 2023) is a Q-learning based algorithm that adds an entropy bonus as a strongly concave term to fix linear scalarization's limitations and recover the Pareto front. **EWP** (Wiltzer et al., 2024) is a distributional extension of CAPQL, where the critic learns value distributions via MMD minimization. **Marginal-IQN** (Zhou et al., 2021) is a multi-objective extension of IQN, without consideration of the dependency structure of objectives. Marginal-IQN (N) is trained by a neutral risk-measure, while Marginal-IQN(RS) is trained by CVaR 50% for the risk-sensitive axis; i.e., $\Phi_i^{-1}(u) = 0.5u_i$ for risk-sensitive objective $i$, and $\Phi_j^{-1}(u) = u$ as described in Eq.(7), for $j = 1, \ldots, d$ and $i \neq j$. **KR-IQN** is our proposed approach. KR-IQN (N) is trained by a neutral risk-measure, while KR-IQN(RS) is trained by CVaR 50% for the risk-sensitive axis.

**Metrics:** We choose four criteria: expected utility (EU), hyper volume (HV), risk-sensitive expected utility $\text{EU}_{\text{risk}}$, and

*Table 1.* Main results. The $i$ in parentheses in Eq.(16) and Eq.(18), as well as in KR-IQN(RS), denotes the index of the objective that is treated as risk-sensitive for each environment. Finance* does not belong to the MO-gymnasium. The bold numbers denote the largest value.

| Environment | Hopper($d = 3, i = 2$) | | | | Cheetah ($i = 2$) | | | |
|---|---|---|---|---|---|---|---|---|
| Algorithm | EU | $EU_{Risk}$ | $HV(\times 10^8)$ | $HV_{Risk}(\times 10^8)$ | EU | $EU_{Risk}$ | $HV(\times 10^6)$ | $HV_{Risk}(\times 10^6)$ |
| PGMORL | $392.45_{\pm 37.58}$ | $376.10_{\pm 50.86}$ | $0.97_{\pm 0.23}$ | $0.87_{\pm 0.31}$ | $-154.66_{\pm 245.19}$ | $-174.78_{\pm 269.10}$ | $1.03_{\pm 0.42}$ | $1.02_{\pm 0.49}$ |
| CAPQL | $796.06_{\pm 211.63}$ | $534.69_{\pm 114.23}$ | $6.05_{\pm 3.15}$ | $5.86_{\pm 3.04}$ | $997.13_{\pm 898.08}$ | $766.38_{\pm 975.55}$ | $7.16_{\pm 3.36}$ | $7.15_{\pm 3.36}$ |
| DPMORL | $988.18_{\pm 56.91}$ | $680.74_{\pm 102.91}$ | $6.59_{\pm 14.24}$ | $3.49_{\pm 1.67}$ | $558.89_{\pm 32.38}$ | $67.44_{\pm 123.78}$ | $4.05_{\pm 0.32}$ | $4.31_{\pm 0.27}$ |
| EWP | $429.75_{\pm 0.56}$ | $376.39_{\pm 48.82}$ | $1.18_{\pm 0.01}$ | $1.18_{\pm 0.01}$ | $1578.34_{\pm 147.46}$ | $1575.96_{\pm 126.87}$ | $9.59_{\pm 0.80}$ | $9.49_{\pm 0.70}$ |
| MarginalIQN (N) | $909.52_{\pm 140.61}$ | $384.95_{\pm 170.54}$ | $7.73_{\pm 2.47}$ | $6.67_{\pm 3.18}$ | $2097.21_{\pm 96.63}$ | $2113.80_{\pm 179.73}$ | $11.48_{\pm 0.36}$ | $11.44_{\pm 0.75}$ |
| MarginalIQN (RS) | $702.57_{\pm 253.62}$ | $181.46_{\pm 84.92}$ | $4.59_{\pm 3.39}$ | $3.10_{\pm 2.74}$ | $1797.59_{\pm 394.39}$ | $1933.87_{\pm 164.57}$ | $10.10_{\pm 1.96}$ | $10.61_{\pm 1.05}$ |
| **KR-IQN(N)** | $1247.39_{\pm 9.55}$ | $934.89_{\pm 232.81}$ | $15.75_{\pm 0.29}$ | $15.33_{\pm 0.51}$ | $\mathbf{2172.71}_{\pm 131.14}$ | $\mathbf{2153.02}_{\pm 139.46}$ | $\mathbf{12.10}_{\pm 0.61}$ | $\mathbf{11.82}_{\pm 0.60}$ |
| **KR-IQN(RS)** | $\mathbf{1253.93}_{\pm 11.36}$ | $\mathbf{1086.62}_{\pm 106.02}$ | $\mathbf{16.28}_{\pm 0.47}$ | $\mathbf{15.96}_{\pm 0.60}$ | $2153.88_{\pm 228.74}$ | $2140.37_{\pm 216.99}$ | $12.02_{\pm 1.01}$ | $11.81_{\pm 0.99}$ |
| Environment | Walker2d ($i = 2$) | | | | Hopper-2d ($i = 1$) | | | |
| Algorithm | EU | $EU_{Risk}$ | $HV(\times 10^5)$ | $HV_{Risk}(\times 10^5)$ | EU | $EU_{Risk}$ | $HV(\times 10^6)$ | $HV_{Risk}(\times 10^6)$ |
| PGMORL | $458.09_{\pm 98.81}$ | $429.47_{\pm 108.92}$ | $4.94_{\pm 1.32}$ | $4.45_{\pm 1.35}$ | $267.78_{\pm 87.23}$ | $225.60_{\pm 113.59}$ | $0.11_{\pm 0.05}$ | $0.08_{\pm 0.06}$ |
| CAPQL | $422.77_{\pm 245.68}$ | $292.41_{\pm 292.74}$ | $4.62_{\pm 3.35}$ | $4.52_{\pm 3.32}$ | $1142.18_{\pm 277.25}$ | $700.93_{\pm 260.20}$ | $1.48_{\pm 0.73}$ | $1.43_{\pm 0.75}$ |
| DPMORL | $953.64_{\pm 42.91}$ | $668.10_{\pm 54.03}$ | $12.82_{\pm 0.65}$ | $9.65_{\pm 2.03}$ | $1583.86_{\pm 34.73}$ | $760.00_{\pm 67.30}$ | $2.80_{\pm 11.73}$ | $2.08_{\pm 0.32}$ |
| EWP | $456.37_{\pm 110.25}$ | $186.76_{\pm 37.24}$ | $4.75_{\pm 1.23}$ | $4.71_{\pm 1.22}$ | $196.53_{\pm 63.37}$ | $108.87_{\pm 54.56}$ | $0.56_{\pm 0.03}$ | $0.39_{\pm 0.02}$ |
| MarginalIQN (N) | $882.81_{\pm 81.82}$ | $606.53_{\pm 131.49}$ | $12.09_{\pm 1.98}$ | $11.25_{\pm 2.08}$ | $1285.86_{\pm 147.01}$ | $641.46_{\pm 262.12}$ | $1.85_{\pm 0.44}$ | $1.65_{\pm 0.58}$ |
| MarginalIQN (RS) | $802.44_{\pm 160.38}$ | $636.37_{\pm 144.19}$ | $10.49_{\pm 2.36}$ | $10.30_{\pm 2.17}$ | $1285.10_{\pm 275.10}$ | $757.54_{\pm 280.67}$ | $1.89_{\pm 0.73}$ | $1.73_{\pm 0.73}$ |
| **KR-IQN(N)** | $883.39_{\pm 156.63}$ | $819.23_{\pm 106.24}$ | $11.32_{\pm 2.37}$ | $11.07_{\pm 2.00}$ | $1572.15_{\pm 330.44}$ | $1337.64_{\pm 535.71}$ | $2.91_{\pm 1.02}$ | $2.78_{\pm 1.20}$ |
| **KR-IQN(RS)** | $\mathbf{1107.29}_{\pm 70.41}$ | $\mathbf{960.98}_{\pm 134.74}$ | $\mathbf{15.88}_{\pm 1.37}$ | $\mathbf{15.72}_{\pm 1.22}$ | $\mathbf{1724.41}_{\pm 12.42}$ | $\mathbf{1675.48}_{\pm 48.84}$ | $\mathbf{3.37}_{\pm 0.04}$ | $\mathbf{3.35}_{\pm 0.03}$ |
| Environment | Ant ($i = 3$) | | | | Finance* ($i = 3$) | | | |
| Algorithm | EU | $EU_{Risk}$ | $HV(\times 10^8)$ | $HV_{Risk}(\times 10^8)$ | EU | $EU_{Risk}$ | $HV(\times 10^6)$ | $HV_{Risk}(\times 10^6)$ |
| PGMORL | $190.61_{\pm 25.95}$ | $207.81_{\pm 28.23}$ | $0.35_{\pm 0.08}$ | $0.21_{\pm 0.12}$ | $-94.30_{\pm 18.52}$ | $-185.66_{\pm 10.06}$ | $5.16_{\pm 0.75}$ | $0.95_{\pm 0.44}$ |
| CAPQL | $577.06_{\pm 121.19}$ | $381.47_{\pm 115.48}$ | $4.63_{\pm 2.47}$ | $4.62_{\pm 2.50}$ | $2.47_{\pm 3.24}$ | $-10.70_{\pm 5.96}$ | $8.52_{\pm 0.98}$ | $9.03_{\pm 0.10}$ |
| DPMORL | $683.85_{\pm 11.20}$ | $602.62_{\pm 35.58}$ | $5.84_{\pm 3.82}$ | $5.85_{\pm 0.53}$ | $13.89_{\pm 5.30}$ | $-63.27_{\pm 29.29}$ | $\mathbf{19.56}_{\pm 2.54}$ | $\mathbf{17.54}_{\pm 1.84}$ |
| EWP | $783.20_{\pm 161.83}$ | $320.84_{\pm 93.87}$ | $9.70_{\pm 4.52}$ | $8.80_{\pm 3.45}$ | $12.67_{\pm 12.38}$ | $2.18_{\pm 8.99}$ | $1.06_{\pm 1.18}$ | $1.07_{\pm 1.26}$ |
| MarginalIQN (N) | $1399.39_{\pm 308.63}$ | $1429.64_{\pm 404.34}$ | $34.86_{\pm 14.21}$ | $35.47_{\pm 12.69}$ | $\mathbf{51.44}_{\pm 6.60}$ | $30.16_{\pm 3.69}$ | $11.59_{\pm 0.80}$ | $12.93_{\pm 0.36}$ |
| MarginalIQN (RS) | $912.61_{\pm 77.73}$ | $948.00_{\pm 101.71}$ | $12.44_{\pm 2.77}$ | $16.95_{\pm 4.21}$ | $50.41_{\pm 7.67}$ | $29.94_{\pm 4.12}$ | $11.68_{\pm 0.55}$ | $12.98_{\pm 0.49}$ |
| **KR-IQN(N)** | $1513.96_{\pm 194.01}$ | $1459.33_{\pm 166.78}$ | $41.81_{\pm 12.41}$ | $39.14_{\pm 7.79}$ | $50.55_{\pm 6.87}$ | $\mathbf{32.05}_{\pm 6.59}$ | $11.49_{\pm 0.46}$ | $13.12_{\pm 0.23}$ |
| **KR-IQN(RS)** | $\mathbf{1596.25}_{\pm 101.15}$ | $\mathbf{1645.08}_{\pm 133.61}$ | $\mathbf{45.87}_{\pm 7.29}$ | $\mathbf{45.69}_{\pm 6.15}$ | $46.36_{\pm 3.92}$ | $27.84_{\pm 5.86}$ | $11.87_{\pm 0.42}$ | $12.99_{\pm 0.37}$ |

risk-sensitive hyper volume $HV_{risk}$. To define the metrics, let $\mathcal{P}^\pi$ be the set of policies generated by MORL algorithms. For a preference-conditioned algorithm, we may set $\mathcal{P}^\pi = \{\pi(\cdot|\omega)|\omega \in \mathcal{W}\}$. Here, $\mathcal{W}$ is the preference set.

Similar to EU and HV, we define $EU_{risk}$ and $HV_{risk}$. $EU_{risk}$ measures the scalarized multi-objective risk. It is defined as

$$\text{EU}_{\text{risk}}(\mathcal{P}^\pi) = \mathbb{E}_{\omega \in \mathcal{W}} \left[ \max_{\pi \in \mathcal{P}^\pi} \mathbb{E}_{\mathbf{Z}^\pi \sim \pi} [\omega^T \rho_{10\%}^{\text{CVaR}}(\mathbf{Z}^\pi; i)] \right], \quad (16)$$

where

$$\rho_{10\%}^{\text{CVaR}}(\mathbf{Z}^\pi; i) = \mathbb{E}[\mathbf{Z}^\pi | \mathbf{Z}_i^\pi \leq F_{\mathbf{Z}_i^\pi}^{-1}(0.1)]. \quad (17)$$

It is the conditional expectation of $\mathbf{Z}^\pi$ when the specific $\mathbf{Z}_i^\pi$ falls within the worst 10% of outcomes. This construction is from a tractable approximation of Eq.(8). The negation is omitted to align with the RL convention of higher is better. $HV_{risk}$ quantifies the risk-adjusted dominated volume in objective space. With a reference point $\mathbf{r}$, it is defined as

$$\text{HV}_{\text{Risk}}(\mathcal{P}^\pi; \mathbf{r}) = \text{Vol}\left( \bigcup_{\pi \in \mathcal{P}^\pi} \prod_{k=1}^d [\rho_{10\%}^{\text{CVaR}}(\mathbf{Z}^\pi; i)_k, \mathbf{r}_k] \right). \quad (18)$$

We evaluate $\text{CVaR}_{10\%}$ instead of $\text{CVaR}_{50\%}$, which is the trained objective, to emphasize the severe failure cases. For the $\rho_{50\%}^{\text{CVaR}}$, see Appendix D.

### 6.1. Main Results

Table 1 depicts the overall results. The proposed method KR-IQN achieves an average performance improvement of 29.77% over the strongest baseline. In particular, on risk-sensitive metrics ($EU_{Risk}$, $HV_{Risk}$), KR-IQN demonstrates an improvement of 41.18%, which is more than twice the improvement observed on standard metrics (18.36%). Excluding the Finance environment, the average improvement reaches 38.72%. In Finance environment, DPMORL outperforms KR-IQN with respect to HV and $HV_{Risk}$. However, in this case, the EU drops substantially, which suggests that DPMORL excessively amplifies a particular objective in a nonlinear manner. This behavior is likely because DPMORL learns multiple policies with respect to randomly sampled nonlinear utility functions. While MarginalIQN (N) shows the largest EU, the difference is negligible considering the standard deviation. We show the details about it in Section 6.2. There are cases where a model trained in a risk-sensitive manner (KR-IQN(RS)) yields a higher

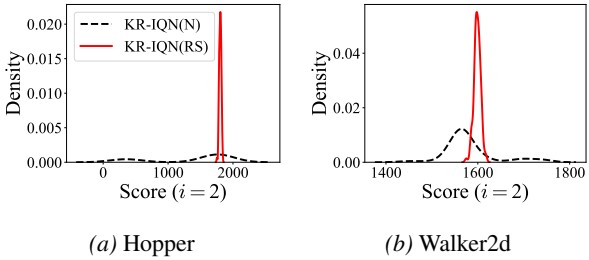

*(a)* Hopper         *(b)* Walker2d

*Figure 4.* Score KDE of Hopper and Walker2d.

risk-neutral metric HV or EU.

KR-IQN (RS) often achieves a higher mean score, the undiscounted reward sum, because it raises the lower tail of the value distribution. This phenomenon has also been observed in (Dabney et al., 2018a); see Figure 4 as well. One may observe that the score of KR-IQN (RS) tends to be concentrated in a higher region. Figure 4 and the graphs in Appendix D.2 provide sufficient evidence for our main claim that KR-IQN (RS) makes risk-sensitive decisions.

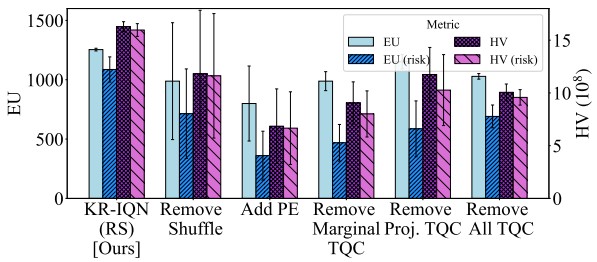

*Figure 5.* Ablation study results (Hopper($d = 3$)).

**Ablation Study:** Figure 5 reports performance degradation relative to the full model. For the ordering perspective, removing shuffle (34.3% ↓ $EU_{risk}$, 27.2% ↓ $HV_{risk}$) and PE (66.9% ↓ $EU_{risk}$, 58.3% ↓ $HV_{risk}$) causes performance drop with higher variance. This suggests that they effectively eliminate unnecessary inductive bias while providing data augmentation. Marginal TQC alone achieves the smallest degradation in expected utility (8.2%↓EU, 27.9%↓HV), but still shows notable drops in risk-sensitive metrics (46.0% ↓ $EU_{risk}$, 35.7% ↓ $HV_{risk}$). Meanwhile, Projected TQC alone performs (56.9% ↓ $EU_{risk}$, 49.8% ↓ $HV_{risk}$) worse than removing both components entirely. This is because a mature MORL policy inherently faces value trade-offs regardless of optimality, inducing negative correlations among values. This aligns with Section 5.2, explaining the degraded performance of projective TQC alone.

### 6.2. Case Study

We aim to investigate how various risk measures affect real-world applications. In particular, for the trading environment in Section 6.1, we use an unseen test interval (Jan 02-Jun 30,

2025) that does not overlap with the training interval. We additionally investigate the Wang and simplex risk measures. The Wang risk measure is defined by the weight function

$$\phi(u; \alpha) = \frac{\varphi(F_{\mathcal{N}(0,1)}^{-1}(1 - u) + \alpha)}{\varphi(F_{\mathcal{N}(0,1)}^{-1}(1 - u))}, \quad (19)$$

in Def. 2. Here, $\varphi$ is the density of the standard normal distribution. As in the case of CVaR, we apply the Wang risk measure only to the third objective. The simplex risk measure is defined in Def. 4.4.

Prioritizing practical applicability over theoretical metrics like EU, we employ Earnings, MDD, Sharpe/Sortino ratios, VaR(5%), and CVaR(10%). We report the average performance of the top-5 policies selected by Sortino ratio up to April 30. The unselected policies ($\mathcal{P}_{\text{Fail}}^{\pi}$) incur an *Incubation Cost* (IC), defined as

$$IC(\mathcal{P}^{\pi}) = \frac{1}{|\mathcal{P}_{\text{Fail}}|} \sum_{\pi \in \mathcal{P}_{\text{Fail}}^{\pi*}} (1 - R_{T_{\text{eval}}}^{\pi*}/R_0^{\pi*}), \quad (20)$$

where $R_t^{\pi}$ is the asset value and $T_{\text{eval}}$ is the selection date. A negative IC implies that even unselected policies yield positive returns. We introduce IC to quantify the cost of run-time policy selection, as methods like DPMORL and PGMORL require evaluation on test data to determine which utility they are optimized for. Detailed definitions of other metrics are provided in the Appendix D.6.

Table 2 summarizes the results. KR-IQN achieves higher earnings, Sortino, and Sharpe ratios compared to both baselines and the benchmark asset. The modest MDD improvement reflects that MDD captures the maximum peak-to-trough drawdown, which cannot align with the memoryless property of (MO-)MDP. Except for the risk-neutral setting, our method yields a negative IC, indicating a pool of highly profitable policies. The risk-neutral setting yields the lowest earnings due to its lack of robustness against distributional shifts, whereas coherent risk measures remain robust via their dual representation $\rho(Z) = \inf_{\mathbb{Q} \in \mathcal{Q}} \mathbb{E}_{\mathbb{Q}}[Z]$, optimizing returns under adverse scenarios.

## 7. Conclusion and Limitation

We introduce KR-IQN, a risk-sensitive MORL framework based on KR quantile regression. We theoretically prove that our framework defines a valid vector-risk measure and yields a convergent MO-Bellman operator. To mitigate KR's inherent artificial ordering, we employ a transformer architecture without PE. We also introduce MO-TQC, which is tailored for MORL to stabilize the framework. Experiments on MO-Gymnasium and financial tasks demonstrate that KR-IQN learns risk-sensitive behaviors.

Although we relax the unnatural ordering induced by the KR map, we cannot eliminate it. The principal way is using

*Table 2.* Finance test data results.

| Category | Name | Earning (%) | Sortino | Sharpe | MDD | $VaR_{5\%}$ | $CVaR_{10\%}$ | IC (%) |
|---|---|---|---|---|---|---|---|---|
| KR-IQN (Ours) | $CVaR_{50\%}$ | $25.40_{\pm21.58}$ | $4.50_{\pm2.99}$ | $2.01_{\pm1.09}$ | $-0.12_{\pm0.07}$ | $23.41_{\pm19.56}$ | $23.32_{\pm19.48}$ | $-2.10_{\pm5.28}$ |
| | Neutral | $18.09_{\pm8.75}$ | $3.98_{\pm2.73}$ | $1.72_{\pm0.84}$ | $-0.17_{\pm0.08}$ | $15.58_{\pm9.66}$ | $15.46_{\pm9.66}$ | $1.04_{\pm3.26}$ |
| | $Simplex_1$ | $27.32_{\pm16.97}$ | $5.30_{\pm3.02}$ | $2.36_{\pm0.99}$ | $-0.15_{\pm0.07}$ | $25.79_{\pm16.02}$ | $25.62_{\pm15.90}$ | $-1.94_{\pm4.26}$ |
| | $Wang_{-0.71}$ | $25.79_{\pm16.28}$ | $5.67_{\pm2.99}$ | $2.40_{\pm0.96}$ | $-0.14_{\pm0.02}$ | $23.39_{\pm15.37}$ | $23.37_{\pm15.28}$ | $-5.35_{\pm5.45}$ |
| Marginal IQN (Baselines) | $CVaR_{50\%}$ | $15.73_{\pm10.65}$ | $3.17_{\pm2.74}$ | $1.55_{\pm1.13}$ | $-0.15_{\pm0.07}$ | $14.58_{\pm10.64}$ | $14.51_{\pm10.63}$ | $-5.04_{\pm8.93}$ |
| | Neutral | $11.08_{\pm4.65}$ | $2.44_{\pm0.78}$ | $1.41_{\pm0.40}$ | $-0.18_{\pm0.04}$ | $10.19_{\pm4.53}$ | $10.15_{\pm4.51}$ | $-0.92_{\pm2.77}$ |
| | $Simplex_{0.5}$ | $13.66_{\pm11.23}$ | $2.84_{\pm3.15}$ | $1.33_{\pm1.13}$ | $-0.18_{\pm0.05}$ | $12.48_{\pm10.35}$ | $12.37_{\pm10.29}$ | $0.10_{\pm7.56}$ |
| | $Wang_{-0.71}$ | $17.23_{\pm13.83}$ | $2.11_{\pm1.26}$ | $1.10_{\pm0.56}$ | $-0.17_{\pm0.07}$ | $15.55_{\pm12.58}$ | $15.38_{\pm12.44}$ | $4.18_{\pm2.73}$ |
| Baselines | EWP | $5.14_{\pm1.71}$ | $1.10_{\pm0.18}$ | $0.65_{\pm0.11}$ | $-0.09_{\pm0.05}$ | $4.90_{\pm1.48}$ | $4.90_{\pm1.48}$ | $1.54_{\pm2.39}$ |
| | PGMORL | $2.77_{\pm1.53}$ | $0.37_{\pm0.21}$ | $0.15_{\pm0.14}$ | $-0.33_{\pm0.03}$ | $-12.69_{\pm2.78}$ | $-14.03_{\pm2.36}$ | $4.52_{\pm0.31}$ |
| | CAPQL | $11.38_{\pm8.27}$ | $2.05_{\pm0.94}$ | $1.18_{\pm0.52}$ | $-0.15_{\pm0.11}$ | $9.53_{\pm6.47}$ | $9.46_{\pm6.41}$ | $-0.53_{\pm3.57}$ |
| | DPMORL | $10.05_{\pm7.36}$ | $1.77_{\pm0.88}$ | $0.93_{\pm0.44}$ | $-0.32_{\pm0.07}$ | $-1.89_{\pm4.69}$ | $-3.01_{\pm4.51}$ | $2.20_{\pm3.28}$ |
| Benchmark Asset | Physical Gold | $22.74$ | $3.81$ | $2.02$ | $-0.07$ | — | — | — |
| | Nasdaq 100 | $7.70$ | $0.88$ | $0.52$ | $-0.23$ | — | — | — |
| | S&P 500 | $5.80$ | $0.70$ | $0.41$ | $-0.19$ | — | — | — |

OT as explained in Section 2. In Appendix B, we discuss why OT fails and outline other possible approaches.

## Impact Statement

This paper presents work whose goal is to advance the field of Machine Learning. There are many potential societal consequences of our work, none of which we feel must be specifically highlighted here.

## Acknowledgement

This work was supported by Institute of Information & communications Technology Planning & Evaluation (IITP) grant funded by the Korea government (MSIT), (RS-2022-II220043, Adaptive Personality for Intelligent Agents, RS-2022-II221045, Self-directed multi-modal Intelligence for solving unknown, open domain problems, RS-2025-02218768, Accelerated Insight Reasoning via Continual Learning, RS-2025-25442569, AI Star Fellowship Support Program (Sungkyunkwan Univ.), RS-2026-25543726, Development of Leading Talent in Medical Domain-Specific Generative AI, RS-2026-25528384, Resource-Intensive AI Technologies Based on Sustainable GPU Integrated Platforms, RS-2019-II190421, Artificial Intelligence Graduate School Program (Sungkyunkwan University)), the National Research Foundation of Korea (NRF) grant funded by the Korea government (MSIT) (No. RS-2026-25474409), IITP-ITRC (Information Technology Research Center) grant funded by the Korea government (MSIT) (IITP-2025-RS-2024-00437633, 10%), IITP-ICT Creative Consilience Program grant funded by the Korea government (MSIT) (IITP-2026-RS-2020-II201821), and by Samsung Electronics Co., Ltd.

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

# A. Proof of Convergence Proposition 4.6

**Proposition 4.6.** *Let* $\mathbf{Z}, \mathbf{Z}' \in \mathcal{Z}$ *be two state-action value distributions in an MO-MDP with countable state and action spaces. Then, for any policy* $\pi$,

$$\bar{d}_\infty(\Pi_{\mathrm{AW}} \mathcal{T}^\pi \mathbf{Z}, \ \Pi_{\mathrm{AW}} \mathcal{T}^\pi \mathbf{Z}') \ \leq \ \gamma \, \bar{d}_\infty(\mathbf{Z}, \mathbf{Z}'), \tag{A.1}$$

*where*

$$\bar{d}_\infty(\mathbf{Z}, \mathbf{Z}') = \sup_{s,a \in \mathcal{S} \times \mathcal{A}} AW_\infty\big(\mathbf{Z}(s,a), \mathbf{Z}'(s,a)\big). \tag{A.2}$$

**Proof of Proposition 4.6:** Applying Lemma A.5 and by the fact that $AW_1(\mathbf{X}, \mathbf{Y}) \leq AW_\infty(\mathbf{X}, \mathbf{Y})$, we obtain the proposition. The first step is defining adapted Wasserstein distance. The second step is verifying that the adapted Wasserstein distance has the same property as the Wasserstein distance. Finally, we repeat the proof of (Dabney et al., 2018b). The detailed processes are as follows.

## A.1. Step 1. Definitions

We define some tools, which are required for the proof. For the convergence analysis, we leverage the adapted Wasserstein (AW-distance). The AW-distance is useful to analyze the behavior of the KR map.

**Definition A.1.** *(Adapted Wasserstein (Beiglböck et al., 2023))* *Let* $\mathbf{X}, \mathbf{Y}$ *be* $d$-*dimensional random vectors. Further, let* $\mu_{\mathbf{X}}, \mu_{\mathbf{Y}}$ *be push-forwarded probability measures on* $\mathbb{R}^d$ *induced from* $\mathbf{X}, \mathbf{Y}$. *Let* $\Gamma(\mu_{\mathbf{X}}, \mu_{\mathbf{Y}})$ *be a set of all couplings of* $\mu_{\mathbf{X}}, \mu_{\mathbf{Y}}$. *The set* $\Gamma_{bc} \subseteq \Gamma$ *of bicausal coupling consists of those* $\mu$ *whose successive disintegration kernels satisfy almost surely,*

$$(\mu_k | \mathbf{X}_{<k}, \mathbf{Y}_{<k}) \in \Gamma((\mu_{\mathbf{X}} | \mathbf{X}_{<k})_k, (\mu_{\mathbf{Y}} | \mathbf{Y}_{<k})_k), \tag{A.3}$$

*i.e., the couplings whose* $k$-*th components only depend on previous* $1, \ldots, k-1$ *components. Further, we define* $p$-*Adapted Wasserstein distance as*

$$AW_p(\mathbf{X}, \mathbf{Y}) = \inf_{\mu \in \Gamma_{bc}} \left( \int_{\mathbb{R}^d} \|\mathbf{X} - \mathbf{Y}\|_p^p d\mu \right)^{\frac{1}{p}}, \tag{A.4}$$

*for* $p \in [0, \infty]$.

Now we extend the quantile projection of Dabney to multi-objective settings.

**Definition A.2.** *Let* $\mathcal{Q}$ *be a set of conditional quantile functions whose codomain's cardinality is fixed* $M \in \mathbb{N}$. *Then,* *AW-projection* $\Pi_{\mathrm{AW}}$ *is defined as*

$$\Pi_{\mathrm{AW}}[\mathbf{Z} | \mathbf{Z}_{<k}]_k = \arg\min_{\mathbf{Q}_\theta \in \mathcal{Q}} W_1(\mathbf{Q}_\theta(\mathbf{u}_k; \mathbf{Z}_{<k}), \mathbf{Q}_{\mathbf{Z}}(\mathbf{u}_k; \mathbf{Z}_{<k})), \tag{A.5}$$

*where* $W_1$ *is the Wasserstein 1-distance. Note that this coincides with* $AW_1$ *from conditionality.*

## A.2. Step 2. $\mathcal{T}^\pi$ is a $\gamma$-contraction:

**Lemma A.3.** *Let*

$$AW_\infty(\mathbf{X}, \mathbf{Y}) := \inf_{cpl(\mathbf{X}, \mathbf{Y}) \in \Gamma_{bc}} \mathrm{ess\,sup} \|\mathbf{X} - \mathbf{Y}\|_\infty. \tag{A.6}$$

*Then* $AW_\infty(\mathbf{X} + \mathbf{R}, \mathbf{Y} + \mathbf{R}) \leq AW_\infty(\mathbf{X}, \mathbf{Y})$, *whenever* $\mathbf{R}$ *is an independent random vector with respect to both* $\mathbf{X}, \mathbf{Y}$. *Note further, it is trivial that* $AW_\infty(a\mathbf{X}, a\mathbf{Y}) = aAW_\infty(\mathbf{X}, \mathbf{Y})$ *for any scalar constant* $a \geq 0$.

**Proof**: For any coupling $cpl \in \Gamma_{\mathrm{bc}}(\mathbf{X}, \mathbf{Y})$, consider the pair of random vector $(\mathbf{X} + \mathbf{Q}_{\mathbf{R}}(\mathbf{u}), \mathbf{Y} + \mathbf{Q}_{\mathbf{R}}(\mathbf{u}))$. Note that $\mathbf{X}_k$ only depends on $\mathbf{X}_{<k}, \mathbf{Y}_{<k}$, and so are $\mathbf{Y}_k, [\mathbf{Q}_{\mathbf{R}}(\mathbf{u})]_k$. By definition of the KR map and independence of $\mathbf{X}$ and $\mathbf{Y}$ by assumption, $(\mathbf{X} + \mathbf{Q}_{\mathbf{R}}(\mathbf{u}), \mathbf{Y} + \mathbf{Q}_{\mathbf{R}}(\mathbf{u}))$ is bicausal. For this,

$$\mathrm{ess\,sup}_{\mathbf{X}, \mathbf{Y} \sim cpl(\mathbf{X}, \mathbf{Y})} \|\mathbf{X} + \mathbf{Q}_{\mathbf{R}}(\mathbf{u}) - \mathbf{Y} - \mathbf{Q}_{\mathbf{R}}(\mathbf{u})\|_\infty = \mathrm{ess\,sup}_{\mathbf{X}, \mathbf{Y} \sim cpl(\mathbf{X}, \mathbf{Y})} \|\mathbf{X} - \mathbf{Y}\|_\infty. \tag{A.7}$$

Since the choice of coupling was arbitrary, we conclude that

$$AW_\infty(\mathbf{X} + \mathbf{R}, \mathbf{Y} + \mathbf{R}) \leq AW_\infty(\mathbf{X}, \mathbf{Y}). \tag{A.8}$$

**Lemma A.4.** *Let $\mathcal{Z}$ be the set of all state-action distributions for some MO-MDP. $\mathcal{T}^{\pi}$ is a $\gamma$-contraction on $(\mathcal{Z}, \bar{d}_{\infty})$, where $\bar{d}_{\infty}$ is defined in Eq.(12) (in Appendix, Eq.(A.2)) for any pair of value distributions $\mathbf{Z}, \mathbf{Z}' \in \mathcal{Z}$.*

**Proof**: From direct calculation and Lemma A.3, we obtain that $\mathcal{T}^{\pi}$ is a $\gamma$-contraction on $(\mathcal{Z}, \bar{d}_{\infty})$:

$$\bar{d}_{\infty}(\mathcal{T}^{\pi}\mathbf{Z}(s,a), \mathcal{T}^{\pi}\mathbf{Z}'(s,a)) = \sup_{(s,a)\in\mathcal{S}\times\mathcal{A}} AW_{\infty}(\mathbf{R}(s,a) + \gamma\mathbf{Z}(s',a'), \mathbf{R}(s,a) + \gamma\mathbf{Z}'(s',a'))$$

$$\leq \gamma AW_{\infty}(\mathbf{Z}(s',a'), \mathbf{Z}'(s',a')) \leq \gamma \sup_{(s,a)\in\mathcal{S}\times\mathcal{A}} AW_{\infty}(\mathbf{Z}(s,a), \mathbf{Z}'(s,a)) = \gamma\bar{d}_{\infty}(\mathbf{Z}(s,a), \mathbf{Z}'(s,a)). \tag{A.9}$$

### A.3. Step 3. $\Pi_{\text{AW}}\mathcal{T}^{\pi}$ is a $\gamma$-contraction.

**Lemma A.5.** *$\Pi_{\text{AW}}\mathcal{T}^{\pi}$ is a $\gamma$-contraction on $(\mathcal{Z}, \bar{d}_{\infty})$.*

Before we prove the Lemma, we need a weak assumption

**Assumption A.6.** *$\mathcal{Z}$ is essentially bounded for any pair of $(s,a) \in \mathcal{S} \times \mathcal{A}$.*

This assumption implies $\gamma < 1$ or finite horizon, and $\mathbf{R}(s,a)$ is essentially bounded. This is a necessary condition for dealing with the value distribution in a plausible way for any RL algorithm. This assumption allows us to fix the upper bound and thereby reduce the proof to the one-dimensional case of Dabney et al. (2018b).

**Proof**: Since $\mathcal{T}^{\pi}$ is a $\gamma$-contraction in $\bar{d}^{\infty}$, it suffices to show the case when $\gamma = 1$. Given a pair of $\mathbf{X}, \mathbf{Y} \in \mathcal{Z}$, let $[\mathbf{X}(s,a; \mathbf{X}_{<k})]_k = \frac{1}{M}\sum_{i=1}^{M}\delta_{\theta_i}^{(k)}(s,a)$, and let $[\mathbf{Y}(s,a; \mathbf{Y}_{<k})]_k = \frac{1}{M}\sum_{i=1}^{M}\delta_{\psi_i}^{(k)}(s,a)$, for some functions $\theta, \psi : \mathcal{S} \times \mathcal{A} \to \mathbb{R}^M$. Further, choose $(k, \mathbf{X}_{<k}, \mathbf{Y}_{<k})$ as a triplet such that

$$(k, \mathbf{X}_{<k}, \mathbf{Y}_{<k}) \in \underset{k, \mathbf{X}_{<k}, \mathbf{Y}_{<k}}{\text{argmax}} \|(\mathbf{X}_k \mid \mathbf{X}_{<k}) - (\mathbf{Y}_k \mid \mathbf{Y}_{<k})\|_{\infty}. \tag{A.10}$$

Here $\|\cdot\|_{\infty}$ is the $L^{\infty}$ norm of a **univariate** random variable. They exist because $\mathbf{X}, \mathbf{Y}$ are bounded random vectors by Assumption A.6, with convention $\mathbf{X}_1|\mathbf{X}_{<1} = \mathbf{X}_1|\emptyset = \mathbf{X}_1$.

Let $\{(s_i, a_i)\}_{i\in I}$ be all the $I$-indexed state-action pairs that are reachable from $(s', a')$ in a single transition, where $I$ is a countable indexing set. Construct a new MDP and a new value distribution for this MDP in which all distributions are given by single Diracs, with a view to applying Lemma A.4. The new MDP is of the following form. We take the state-action pair $(s', a')$ and define new states, actions, transitions, and policy $\tilde{\pi}$, so that state-action pairs reachable from $(s', a')$ in this new MDP are given by $\{(\tilde{s}_i^{(j)}, \tilde{a}_i^{(j)})\}_{i\in I}^{j=1,\dots,M}$, and probability of reaching the state-action pair $(\tilde{s}_i^{(j)}, \tilde{a}_i^{(j)})$ is $p_i/M$. Further, we define new value distributions $\tilde{\mathbf{X}}, \tilde{\mathbf{Y}}$ as follows. For each $i \in I$ and $j \in \{1, \dots, M\}$, we set:

$$\tilde{\mathbf{X}}(\tilde{s}_i^{(j)}, \tilde{a}_i^{(j)}) = \delta_{\theta_j}^{(k)}(s_i, a_i), \tilde{\mathbf{Y}}(\tilde{s}_i^{(j)}, \tilde{a}_i^{(j)}) = \delta_{\psi_j}^{(k)}(s_i, a_i). \tag{A.11}$$

By construction of AW-projection and Assumption A.6, the difference $|\delta_{\theta_j}^{(k)}(s_i, a_i) - \delta_{\psi_j}^{(k)}(s_i, a_i)|$ is bounded by $AW_{\infty}$. Therefore, the following holds:

$$\bar{d}_{\infty}([\Pi_{\text{AW}}\mathcal{T}^{\tilde{\pi}}\tilde{\mathbf{X}}(s,a)]_{\leq d}, [\Pi_{\text{AW}}\mathcal{T}^{\tilde{\pi}}\tilde{\mathbf{Y}}(s,a)]_{\leq d}) \leq \sup_{\substack{i\in I \\ j=1,\dots,M}} |\delta_{\theta_j}^{(k)}(s_i, a_i) - \delta_{\psi_j}^{(k)}(s_i, a_i)|$$

$$\leq \sup_{i\in I} AW_{\infty}(\mathbf{X}(s_i, a_i; \mathbf{X}_{<k}), \mathbf{Y}(s_i, a_i; \mathbf{Y}_{<k})). \tag{A.12}$$

By choice of $k$, $\mathbf{X}_{<k}$, and $\mathbf{Y}_{<k}$, we have

$$\sup_{i\in I} AW_{\infty}(\mathbf{X}(s_i, a_i; \mathbf{X}_{<k}), \mathbf{Y}(s_i, a_i; \mathbf{Y}_{<k})) = \bar{d}_{\infty}(\mathbf{X}(s_i, a_i), \mathbf{Y}(s_i, a_i)), \tag{A.13}$$

as desired.

## B. Theoretical Discussion

### B.1. MMD Failure of (Adapted) Wasserstein Distance Minimization

**Proposition 1. of Dabney et al. (2018b)** *Let $Z_{\theta}$ be a quantile distribution, and $\hat{Z}_m$ be the empirical distribution composed of $m$ samples from Z. Then for all $p \geq 1$, there exists a $Z$ such that*

$$\text{argmin}\,\mathbb{E}[W_p(\hat{Z}_m, Z_{\theta})] \neq \text{argmin}\,W_p(Z, Z_{\theta}) \tag{B.14}$$

**Corollary B.1.** *(MMD Failure) Let $\mathbf{Z}_\theta$ be a $d$-dimensional random vector, and $\mathbf{Z}_m$ be the empirical distribution composed of $m$ samples from Z. Then for all $p \geq 1$, there exists a $\mathbf{Z}$ such that*

$$\mathrm{argmin}\,\mathbb{E}[AW_p(\hat{\mathbf{Z}}_m, \mathbf{Z}_\theta)] \neq \mathrm{argmin}\,AW_p(\mathbf{Z}, \mathbf{Z}_\theta) \tag{B.15}$$

*and so is for the $W_p$.*

**Proof:** Set $d = 1$. Then, $\mathbf{Z}_\theta$ is a quantile distribution by the pushforward property. From Proposition 1. of Dabney et al. (2018b), the statement holds. For $d \geq 2$, consider the almost ideal case $[\mathbf{Z}_\theta]_{<d} \sim \mathrm{Law}(\mathbf{Z}_{<d})$. By choice of $\mathbf{Z}$, we only consider a random variable $\mathbf{Z}_d | \mathbf{Z}_{<d}$. However, from the case when $d = 1$, this fails and Corollary B.1 holds for $AW_p$. For the $W_p$, consider the case $Z_d$ is independent of $Z_{<d}$, then for the case when $W_p$, Corollary B.1 holds.

Proposition 1 of Dabney et al. (2018b) implies not only the failure of MMD but also the potential failure of naive optimal transport implementations. Since RL is fundamentally based on TD or explicit return-based learning, it is inherently a sample-based learning paradigm. Therefore, the fact that the sample Wasserstein minimizer does not converge to the minimizer of the Wasserstein distance with respect to the true distribution is a critical limitation. This implies that, at least for naive OT-based methods, the theoretical utility is severely limited relative to the substantial computational cost (batch OT-based methods using the Hungarian algorithm require at least $\mathcal{O}(BdN^3C)$, where $B$ is the batch size, $d$ is the number of objectives, $N$ is the number of samples, and $C$ is the number of critics).

In contrast, KR-IQN has a computational complexity of $\mathcal{O}(BdNC\log(dNC) + Bd^2NC)$ (the $d^2$ term arises from the transformer architecture, and the sorting operation from TQC costs approximately $dNC\log(dNC)$). Here, $d$ and $C$ are typically much smaller than the number of samples $N$ (among the well-known benchmarks in MO-Gymnasium (Felten et al., 2023), only Fruit Tree has $d \geq 5$), and since sorting is a well-known parallelizable operation, the practical cost is even lower.

### B.2. Discussion about Multivariate Quantile

By definition of spectral risk measure, extending a quantile function from the univariate case to the multivariate case is important to define and calculate the risk. First, for clarity, we review the definition and properties of the univariate quantile. Then, we discuss other approaches and their limitations.

**Review of Univariate Quantile and Quantile Regression Loss:** We may rewrite the quantile regression loss Eq.(3) (including $u \in (0, 1)$ as a parameter of the loss function) as follows:

$$\mathcal{L}_{QR}(\theta; Z, u) = \begin{cases} u|Z - \theta| & \text{if } \theta - Z \geq 0 \\ (1 - u)|Z - \theta| & \text{otherwise.} \end{cases} \tag{B.16}$$

Here, $u$ and $(1 - u)$ can be interpreted as the probability of data (realized random variable) lying below and above the quantile $q$, respectively. That is, the $u$-quantile partitions the data such that a fraction $u$ lies below and $(1 - u)$ lies above. In other words, it is an asymmetric $L^1$ loss according to the $u$ (Koenker, 2005). Focusing on this property, the directional quantile according to the Tukey depth is induced.

Meanwhile, we may rewrite the quantile regression loss as follows:

$$\mathcal{L}_{QR}(\theta; Z, u) = \frac{|Z - \theta| + (2u - 1)(Z - \theta)}{2}. \tag{B.17}$$

Subtracting the constant at $\theta = 0$ from Eq.(B.17) yields

$$\mathcal{L}_{\text{Centered } QR}(\theta; Z, u) := \mathcal{L}_{QR}(\theta; Z, u) - \mathcal{L}_{QR}(0; Z, u) = \frac{|Z - \theta| + (2u - 1)(Z - \theta)}{2} - \frac{|Z| + (2u - 1)Z}{2}. \tag{B.18}$$

Since $\mathcal{L}(0; Z, u)$ is a constant for fixed $Z$ and $u$, the minimizer is invariant with respect to translation. Eq.(B.18) induces the spatial quantile.

**Spatial Quantile (Abdous & Theodorescu, 1992):** The spatial quantile is a definition induced by the loss function form of Eq.(B.18). For a more intuitive explanation, we examine the median case argument proposed by Kemperman (1987). The median of a random variable $Z$ can be written as

$$\mathrm{argmin}_{\theta \in \mathbb{R}} \mathbb{E}[|Z - \theta|]. \tag{B.19}$$

Extending this to a $d$-dimensional random vector $\mathbf{Z}$, we have median of random vector under assumption that $\mathbb{E}[\|\mathbf{Z}\|_p] < \infty$

$$\mathrm{argmin}_{\theta \in \mathbb{R}^d} \mathbb{E}[\|\mathbf{Z} - \theta\|_p] \tag{B.20}$$

To avoid the additional assumption that expectation of $\|\mathbf{Z}\|_p$ is finite, Eq.(B.20) can be written as

$$\mathrm{argmin}_{\theta \in \mathbb{R}^d} \mathbb{E}[\|\mathbf{Z} - \theta\|_p - \|Z\|_p] \tag{B.21}$$

This is the motivation for defining spatial quantile and $\mathcal{L}_{\text{Centered } QR}$.

Following the definition of multivariate median of Kemperman (1987), one may (re-)define the $u$-quantile as a minimizer of (centred) quantile regression loss;

$$\mathrm{argmin}_{\theta \in \mathbb{R}} \mathcal{L}_{\text{Centered } QR}(\theta; Z, u). \tag{B.22}$$

**Definition B.2.** *The spatial quantile $\mathbf{Q}_{\mathbf{Z};p}^{Spatial} : [0,1] \to \mathcal{B}(\mathbb{R}^d)$ is a set-valued map, whose function value is containing a minimizer of the $d$-dimensional extended $\mathcal{L}_{CenteredQR}$*

$$\mathbf{Q}_{\mathbf{Z};p}^{Spatial}(u) = \{\mathbf{x} \in \mathbb{R}^d | \mathbf{x} = argmin_\theta \mathbb{E}[\|\mathbf{Z} - \theta\|_{p;u} - \|\mathbf{Z}\|_{p;u}]\}, \tag{B.23}$$

*where*

$$\|\mathbf{x}\|_{p;u} = \left\| \frac{\mathbf{x}_1 + (2u-1)\mathbf{x}_1}{2}, \dots, \frac{\mathbf{x}_d + (2u-1)\mathbf{x}_d}{2} \right\|_p. \tag{B.24}$$

Eq.(B.24) is followed from $\theta$ dependent term of Eq.(B.18). Not only is the spatial quantile a set-valued map, but it also cannot exaggerate the influence of each individual component when calculating risk.

**Tukey Depth and Directional Quantile (Hallin et al., 2010; Kong & Mizera, 2012):** Tukey depth focuses on the partitioning property of the univariate quantile. It measures how far a data point lies from the center of a distribution. The definition of Tukey depth $\mathcal{D}_{\text{Tukey}}$ is as follows:

$$\mathcal{D}_{\text{Tukey}}(\mathbf{x}; \text{Law}_{\mathbf{X}}) = \inf_{\mathbf{v} \in \mathbb{S}^{d-1}} \Pr[\mathbf{v}^T(\mathbf{x} - \mathbf{X}) \geq 0], \tag{B.25}$$

where $\mathbf{x} \in \mathbb{R}^d$ is a real vector, $\mathbf{X}$ is a $d$-dimensional random vector, and $\text{Law}_{\mathbf{X}}$ denotes the probability distribution of $\mathbf{X}$. Eq.(B.25) corresponds to the Eq.(B.16). This allows us to extend univariate quantile regression and derive a directional quantile loss that incorporates Tukey depth.

This partitioning property naturally extends to the multivariate setting by considering projections onto each direction $\mathbf{v} \in \mathbb{S}^{d-1}$. For a given direction $\mathbf{v}$, we project the random vector $\mathbf{X}$ onto $\mathbf{v}$ and perform univariate quantile regression. Hallin et al. (Hallin et al., 2010) formalized this idea: the directional index $\boldsymbol{\tau} \in B^d := \{\mathbf{x} \in \mathbb{R}^d : 0 < \|\mathbf{x}\| < 1\}$ factorizes into $\boldsymbol{\tau} = \|\boldsymbol{\tau}\|\mathbf{v}$[5]. Let $V_{\mathbf{v}} \in \mathcal{M}_{d \times (d-1)}(\mathbb{R})$ be a $d \times (d-1)$ matrix whose columns form an orthonormal basis for the orthogonal complement of $\mathbf{v}$.

**Definition B.3.** *The directional quantile $\mathbf{Q}_{\mathbf{Z}}^{dir.}$ is a set-valued map*

$$\mathbf{Q}_{\mathbf{Z}}^{dir.}(\boldsymbol{\tau}) = \left\{ \{\mathbf{x} \in \mathbb{R}^d \mid \mathbf{v}^T\mathbf{x} = \mathbf{b}_{\boldsymbol{\tau}}^T V_{\mathbf{v}}^T \mathbf{x} + a_{\boldsymbol{\tau}}\} | a_{\boldsymbol{\tau}}, \mathbf{b}_{\boldsymbol{\tau}} = argmin_{(a,\mathbf{b})} \mathbb{E}\left[ chk_{\|\boldsymbol{\tau}\|}(\mathbf{v}^T\mathbf{Z} - \mathbf{b}^T V_{\mathbf{v}}^T \mathbf{Z} - a) \right] \right\}, \tag{B.26}$$

*where*

$$chk_u(x) := x(u - \mathbb{I}(x < 0))^{[6]} \tag{B.27}$$

Also recall that $Q(\theta; Z, u) = \text{chk}_u(|Z - \theta|)$, which connects to univariate quantile regression.

Both spatial quantile and directional quantile are set-valued maps, making them unsuitable for deep RL. In contrast, optimal transport is vector-valued, which yields more favorable properties.

**Optimal Transport (Carlier et al., 2016):**

---

[5]Here, we introduce the variable $\boldsymbol{\tau}$ instead of $\mathbf{u}$ to emphasize that their domains are different.

[6]This is an abbreviation for the check function, usually denoted as $\rho_\tau$. Since we already use $\rho$ as a risk measure, we use chk instead of $\rho$.

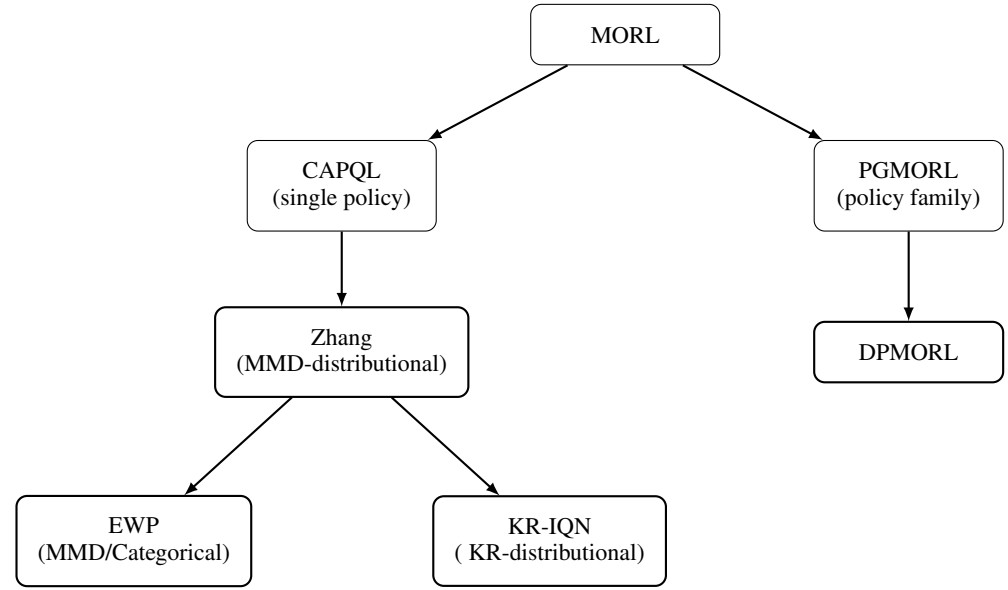

*Figure A.* Taxonomy of MORL methods.

**Definition B.4.** *A multivariate quantile of a random vector* $\mathbf{X}$ *defined by optimal transport (Carlier et al., 2016) is a map* $\mathbf{Q}_{\mathbf{X}}^{OT} : [0,1]^d \to \mathbb{R}^d$, *where*

$$\mathbf{Q}_{\mathbf{X}}^{OT} = \underset{T:T_{\#}U[0,1]^d \cong \mathbf{X}}{arginf} \mathbb{E}_{\mathbf{u} \sim U[0,1]^d} \left[ c(\mathbf{u}, T(\mathbf{u})) \right], \tag{B.28}$$

*where* $c : [0,1]^d \times \mathbb{R}^d \to [0,\infty]$ *is a cost function, which is usually chosen to be the* $L^2$ *norm.*

The Eq.(B.28) is known as the Monge problem, and finding a tractable solution has been considered extremely difficult (Villani et al., 2008). To circumvent this, one may either solve the Kantorovich problem and derive a map from it, or directly find the Brenier potential; however, both approaches remain highly challenging (Villani et al., 2008). A relatively indirect approach is solving the Kantorovich-Rubinstein duality, as used in WGAN (Arjovsky et al., 2017), but this only minimizes the Wasserstein distance between the learned and generated distributions, i.e., it only produces a pushforward and does not directly solve the OT problem.

Meanwhile, even if one manages to solve it, the discussion in Corollary B.1 still applies to the OT-based approach. Combining Tukey depth or spatial quantiles with optimal transport, instead of the proposed KR map, may yield promising results; however, this remains beyond the scope of this paper.

However, one notable advantage is that the source distribution in OT maps is not restricted to be uniform. Much like specifying a prior in Bayesian inference, one can incorporate domain knowledge by choosing an informative source distribution, which is similar to copula-based methods (Bernard et al., 2014).

For example one may choose a Gaussian distribution $\mathcal{N}(\boldsymbol{\mu}, \boldsymbol{\Sigma})$ as a source, where $\boldsymbol{\mu} \in \mathbb{R}^d$ and $\boldsymbol{\Sigma} \in \mathcal{M}(\mathbb{R})_{d \times d}$ is a covariance matrix. This does not conflict with the uniform-based formulation: extending the codomain to $\bar{\mathbb{R}}^d$ allows a map $([0,1]^d, U) \xmapsto{\text{OT}} (\bar{\mathbb{R}}^d, \mathcal{N}) \xmapsto{\text{OT}} (\bar{\mathbb{R}}^d, \nu)$, where $\bar{\mathbb{R}} = \mathbb{R} \cup \{-\infty, \infty\}$ and $\nu$ is the target distribution. Since there is no degenerate set of positive measure (i.e., atom) under $U$ or $\mathcal{N}$ measure, the OT map $([0,1]^d, U) \xmapsto{\text{OT}} (\bar{\mathbb{R}}^d, \mathcal{N})$ is a measure-preserving bijection almost everywhere.

## C. Taxonomy of MORL Literature

Zhang et al. (2021) introduced the first MMD-based distributional MORL algorithm. This work was extended by Wiltzer et al. (2024), who theoretically formalized the multi-objective distributional Bellman equation and proposed corresponding particle-based and categorical algorithms.

While MMD-based methods lack the transport property, which is necessary for calculating risk, categorical methods are

*Table A.* Hyperparameters of KR-IQN/Marginal IQN/EWP

| Item | MO-gymnasium | Finance |
|---|---|---|
| Optimizer | Adam ($\beta_1 = 0.5$, $\beta_2 = 0.9$) | |
| Learning rate | $3 \times 10^{-4}$ | |
| Batch size | 256 | |
| $\gamma$ | 0.99 | |
| Entropy coef. | Auto | $5 \times 10^{-2}$ |
| Replay buffer | $10^6$ | |
| Target update | $5 \times 10^{-3}$ | |
| Num. critics | 3 | |
| Training Steps. | $1 \times 10^6$ | $7.5 \times 10^4$ |
| Truncation Lower (Marginal TQC) | 0 (Hopper, 1) | 1 |
| Truncation Upper (Marginal TQC) | 2 | 2 |
| Num. quantiles per objective (critic training) | 16 | |
| Num. quantiles per objective (actor training) | 8 | |

*Table B.* Hyperparameters of PGMORL

| Item | Value |
|---|---|
| Base algorithm | PPO |
| Optimizer | Adam |
| Learning rate | $3 \times 10^{-4}$ |
| Batch size | 32 |
| $\gamma$ | 0.995 (MO-gymnasium) |
| | 0.99 (Finance) |
| | 4 (Finance) |
| Num. Environments | 6 (Cheetah, Hopper-2d, Walker2d) |
| | 15 (Hopper, Ant) |
| | 80 (Cheetah, Walker2d) |
| Warmup Iterations | 80 (Finance) |
| | 200 (Hopper-2d, Hopper, Ant) |
| | 20 (Cheetah, Finance) |
| Evolutionary Iterations | 100 (Hopper-2d, Walker2d) |
| | 210 (Hopper, Ant) |
| Performance buffer size | 2 |
| Num. of weight candidates | 7 |
| Sparsity coefficient | $-1.0$ (Finance, Cheetah, Hopper-2d, Walker2d) |
| | $-1 \times 10^6$ (Hopper, Ant) |
| Training Steps | $1 \times 10^6$ (MO-gymnasium) |
| | $7.4 \times 10^5$ (Finance) |

*Table C.* Hyperparameters of CAPQL

| Item | Value |
|---|---|
| Optimizer | Adam |
| Learning rate | $3 \times 10^{-4}$ |
| Batch size | 256 |
| $\gamma$ | 0.99 |
| Buffer Size | $1 \times 10^6$ |
| Target smoothing coefficient ($\tau$) | 0.005 |
| | 0.2 (Hopper, Hopper-2d) |
| Augmentation strength ($\alpha$) | 0.1 (Cheetah, Ant) |
| | 0.05 (Walker2d, Finance) |
| Training Steps | $1 \times 10^6$ ($7.5 \times 10^4$, Finance) |

*Table D.* Hyperparameters of DPMORL

| Item | Value |
|---|---|
| Base algorithm | PPO |
| Optimizer | Adam |
| Learning rate | $3 \times 10^{-4}$ |
| Batch size | 256 |
| $\gamma$ | 0.99 |
| Num. policies | 20 (Finance 15) |
| Training Steps | $1 \times 10^7$ ($7.4 \times 10^5$, Finance) |

constrained by the need for predefined supports (i.e., domain knowledge). Given this limitation, we exclude the categorical approach and benchmark our method primarily against the MMD approach. (Cai et al., 2023) proposed a framework which learns predefined non-decreasing utility functions, DPMORL.

Cai et al. (2023) introduced DPMORL, which learns policies optimized for predefined non-decreasing utility functions based on the principle of first-order stochastic dominance. Although theoretically robust when all possible utilities are considered, practical constraints limit the method to a few dozen of learned policies. This creates a scalability issue: as the objective dimension grows, a handful of policies fails to adequately cover the preference space. Röpke et al. (2023) provide a theoretical discussion on these limitations.

Meanwhile, Zhou et al. (2021) and Kim et al. (2023) proposed marginal-based distributional RL frameworks, with Kim et al. (2023) specifically targeting Constrained MDPs. However, as discussed earlier, these marginal-based approaches inherently fail to capture the dependencies among distinct objectives.

In non-distributional MORL, Yang et al. (2019) introduced the preference-conditioned framework in non-distributional MORL. Building on this, Lu et al. (2023) proposed CAPQL, showing that stochastic policies with entropy maximization are superior for approximating Pareto frontiers; our base implementation follows this approach. In contrast, Xu et al. (2020) proposed PGMORL, an evolutionary algorithm that generates a family of policies. While PGMORL is the foundational basis for DPMORL, our work diverges by focusing on a single preference-conditioned policy rather than a policy family.

Figure A outlines the historical context of the field and the position of our proposed approach.

## D. Experiment Detail

### D.1. Hyperparameters

Tables A, B, C, and D summarize the hyperparameters used in our experiments. For DPMORL, we followed the original implementation (Zhang, 2023), which uses Stable-baselines3 (Raffin et al., 2021). To ensure version compatibility and

reduce wall-clock time, we modified the implementation to use the JAX-based version (Raffin, 2022). PGMORL and CAPQL were implemented using (Felten et al., 2023), and we adopted the hyperparameters specified in the original papers (Lu et al., 2023; Xu et al., 2020) when available.

Since KR-IQN, Marginal IQN, and EWP share a similar base architecture, we used the same network structure and only differentiated the critic head, which captures the essential differences among the methods. Note that KR-IQN required minimal hyperparameter tuning, implying its robustness to hyperparameter choices.

### D.2. MO-Gymnasium

For MO-Gymnasium, we did not perform any additional engineering. We only fixed the maximum time step to 500 to align with other works in the literature. When computed using marginal risk with 50% CVaR, KR-IQN (RS) achieved the best performance across all environments except for Cheetah and Finance. For Cheetah, the performance is nearly identical, so we cannot conclude that it is statistically worse. Furthermore, for Finance, KR-IQN (N) and KR-IQN (RS) achieve the highest EU$_{\text{Risk}}$ when computed at 50%.

The risk-sensitive objectives for Walker2d, Ant, and Cheetah are centered on energy, as described in Section 1. In contrast, Hopper uses z-axis height ($i = 2$) and Hopper2d uses x-axis velocity ($i = 1$) as their respective risk-sensitive objectives. For Hopper, this design aligns with the inherent risk of episode termination when the agent loses balance due to the locomotion characteristics. These settings represent the problem of optimizing risk in different directions under different objectives within the same environment.

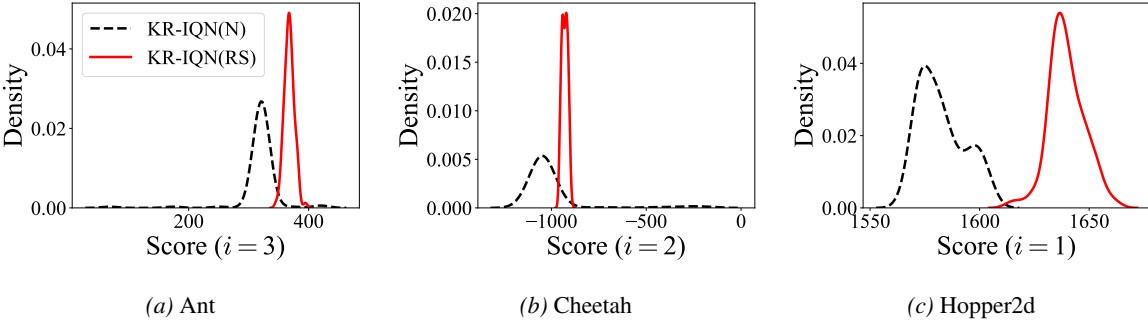

|     (a) Ant     |     (b) Cheetah     |     (c) Hopper2d     |

*Figure B.* Score KDE plot for other MO-gymnasium environment

Investigating multiple risk-sensitive objectives and conducting extensive experiments representing different risks within the same environment could be valuable. However, we did not pursue the former due to substantial computational cost, and the latter is already represented by Hopper. See Appendix E for further discussion.

Figure B presents KDE plots of scores for the risk-sensitive objective in environments not included in the main text. KR-IQN (RS) exhibited more risk-sensitive behavior, which provides sufficient evidence to support our main claim that KR-IQN (RS) makes more risk-sensitive decisions.

Figure C depicts the learning curves of our algorithm. Because $\omega \in \triangle$ is chosen at random, the curves exhibit some variation. Furthermore, there is an inevitable oscillation in the learning dynamics; for more details regarding this phenomenon, see Section 4.1 of (Stooke et al., 2020); Stooke et al. addressed the difficulties caused by the Lagrangian, but the underlying dynamics remain essentially unchanged in the context of MORL.

### D.3. Finance

**Observation:** To implement the financial environment, we slightly modified the FinRL framework. The components of observation are described in Table G. All technical indicators are shifted by one time step to prevent data leakage. For preprocessing, price-related features (OHLC, Volume, EMA) are log-transformed and then standardized, while other technical indicators (RSI, CCI, MACD) are only standardized. The standardization parameters (mean and standard deviation) are computed solely from the training data to avoid information leakage. For the Buy Avg./ Open price feature, we apply the transformation $x \mapsto \log(1 + x)$ for preprocessing. For RSI and CCI, we use periods of 7 and 14, while EMA is calculated with periods of 5, 10, 20, 50, and 100. Finally, for the number of shares, we apply $x \mapsto (x/100) - 3$ for preprocessing. The

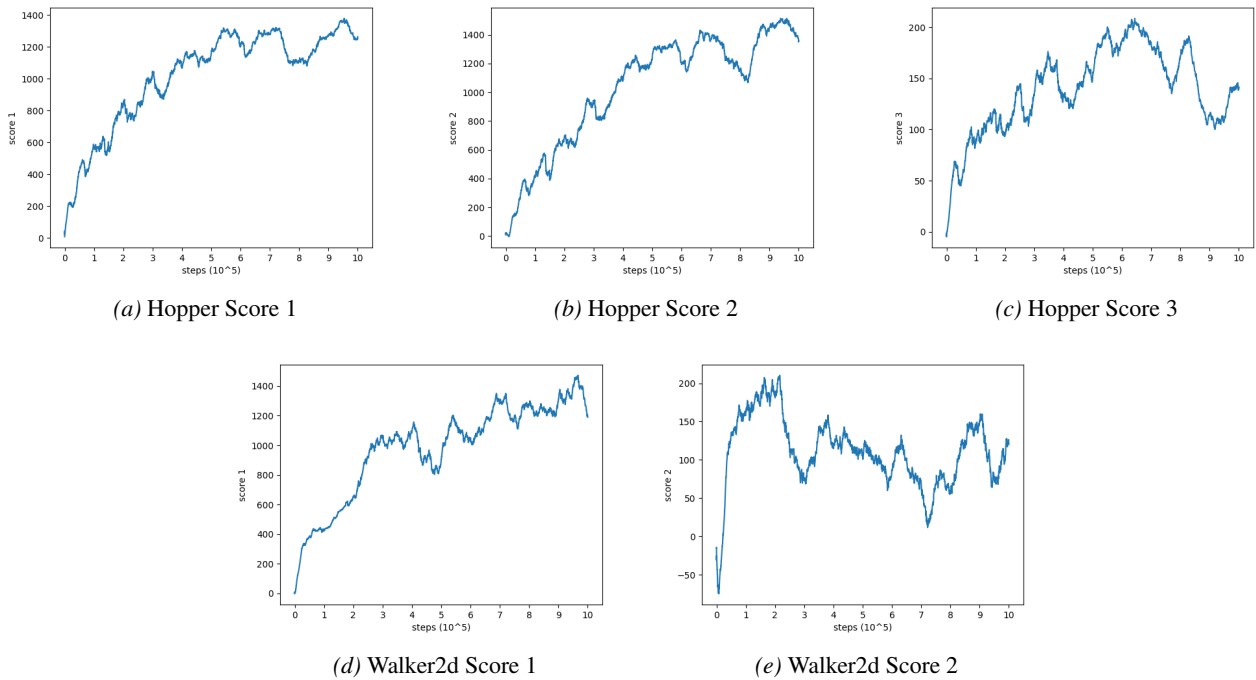

*(a)* Hopper Score 1         *(b)* Hopper Score 2         *(c)* Hopper Score 3

*(d)* Walker2d Score 1         *(e)* Walker2d Score 2

*Figure C.* Learning curves examples

training data spans from May 1, 2019 to December 30, 2024, while the test data covers January 2, 2025 to June 30, 2025.

**Transition:** For bidding prices, we use adverse uniform sampling during training, where the price is uniformly sampled between the $U[\text{Low}, \min(\text{Open}, \text{Close})]]$. During testing, the bidding price is uniformly sampled between the $U[\text{Low}, \text{High}]$. This bidding design assumes that a separate bidding policy exists, but its quality is unknown. The more challenging bidding method during training is intentionally chosen to mitigate overfitting to some extent without relying on additional training techniques. Both buy and sell transactions incur a 0.1% transaction cost. Additionally, an automatic stop-loss is triggered when an asset's price drops below 15% of its average purchase price.

**Multi-objective:** The rewards are defined according to the following principles:

1. Lower volatility
2. Asset distribution: How uniformly are the assets distributed?
3. Logarithmic earnings

**Reward 1 (Lower Volatility):** The first reward is defined as

$$R_1(s, a) = 10^{-4} \cdot \min(\text{Asset}_{t+1} - \text{Asset}_t, 0), \tag{D.29}$$

where $\text{Asset}_t$ is the sum of all assets defined above at time-step $t$, and $10^{-4}$ is a scaling coefficient. This reward penalizes only negative returns, thereby reducing downside volatility.

**Reward 2 (Asset Distribution):** The second reward is defined as,

$$R_2(s, a) = 0.1 \cdot \left( -\sum_{i=1}^{4} \text{NormedAsset}_i \log[\text{NormedAsset}_i] - 0.5 \right), \tag{D.30}$$

i.e., the entropy of the asset distribution with a constant offset. Here, $\text{NormedAsset}_i = \frac{\text{asset}_i}{\sum_{j=1}^{4} \text{asset}_j}$, and

1. $\text{asset}_1$ is the commodity: XLE (Energy Sector Equity[7] ETF), GLD (Physical gold), SLV (Physical silver),

---

[7]Non-precious metal commodities such as crude oil or natural gas are typically traded via futures-based ETFs/ETNs, which exhibit price distortions due to contango and backwardation. Such distortions can adversely affect model training, so we use Energy Sector Equity ETF as a proxy for commodity exposure.

*Table E.* Main Table 1 results evaluated by $\rho_{50\%}^{\text{CVaR}}$ $\text{EU}_{\text{Risk}}$ and $\text{HV}_{\text{Risk}}$.

| Environment | Cheetah ($i=2$) | | Walker2d ($i=2$) | | Hopper-2d ($i=1$) | |
|---|---|---|---|---|---|---|
| Algorithm | $\text{EU}_{\text{Risk}}$ | $\text{HV}_{\text{Risk}}(\times 10^6)$ | $\text{EU}_{\text{Risk}}$ | $\text{HV}_{\text{Risk}}(\times 10^5)$ | $\text{EU}_{\text{Risk}}$ | $\text{HV}_{\text{Risk}}(\times 10^6)$ |
| PGMORL | $-161.34_{\pm255.36}$ | $1.04_{\pm0.45}$ | $446.47_{\pm106.07}$ | $4.76_{\pm1.37}$ | $249.18_{\pm97.07}$ | $0.09_{\pm0.06}$ |
| CAPQL | $863.43_{\pm1004.38}$ | $7.16_{\pm3.37}$ | $331.87_{\pm268.84}$ | $4.56_{\pm3.36}$ | $795.82_{\pm265.50}$ | $1.46_{\pm0.73}$ |
| DPMORL | $43.06_{\pm116.03}$ | $4.28_{\pm0.27}$ | $661.47_{\pm35.34}$ | $11.76_{\pm1.01}$ | $1150.44_{\pm71.59}$ | $2.62_{\pm0.14}$ |
| EWP | $1625.17_{\pm125.51}$ | $9.64_{\pm0.75}$ | $198.69_{\pm25.67}$ | $4.74_{\pm1.23}$ | $128.33_{\pm61.48}$ | $0.05_{\pm0.02}$ |
| MarginalIQN (N) | $\mathbf{2197.77}_{\pm174.34}$ | $11.65_{\pm0.83}$ | $646.27_{\pm157.10}$ | $11.72_{\pm2.09}$ | $811.85_{\pm305.17}$ | $1.74_{\pm0.51}$ |
| MarginalIQN (RS) | $2034.79_{\pm182.78}$ | $10.38_{\pm1.87}$ | $661.23_{\pm147.57}$ | $10.07_{\pm2.10}$ | $866.89_{\pm324.21}$ | $1.84_{\pm0.74}$ |
| **KR-IQN(N)** | $2197.15_{\pm142.30}$ | $\mathbf{12.01}_{\pm0.62}$ | $840.80_{\pm110.75}$ | $11.22_{\pm2.14}$ | $1368.30_{\pm509.23}$ | $2.80_{\pm1.21}$ |
| **KR-IQN(RS)** | $2176.83_{\pm221.50}$ | $11.94_{\pm1.00}$ | $\mathbf{1019.89}_{\pm146.20}$ | $\mathbf{15.60}_{\pm1.07}$ | $\mathbf{1724.71}_{\pm16.84}$ | $\mathbf{3.36}_{\pm0.04}$ |
| Environment | Ant ($i=3$) | | Hopper ($i=2$) | | Finance* ($i=3$) | |
| Algorithm | $\text{EU}_{\text{Risk}}$ | $\text{HV}_{\text{Risk}}(\times 10^8)$ | $\text{EU}_{\text{Risk}}$ | $\text{HV}_{\text{Risk}}(\times 10^8)$ | $\text{EU}_{\text{Risk}}$ | $\text{HV}_{\text{Risk}}(\times 10^6)$ |
| PGMORL | $214.01_{\pm22.23}$ | $0.30_{\pm0.12}$ | $385.27_{\pm43.75}$ | $0.93_{\pm0.26}$ | $-135.46_{\pm24.02}$ | $3.13_{\pm0.99}$ |
| CAPQL | $425.65_{\pm133.48}$ | $4.64_{\pm2.48}$ | $572.08_{\pm137.38}$ | $6.04_{\pm3.16}$ | $3.85_{\pm2.34}$ | $9.27_{\pm0.80}$ |
| DPMORL | $656.85_{\pm29.69}$ | $6.91_{\pm0.56}$ | $642.83_{\pm88.97}$ | $5.72_{\pm1.51}$ | $-33.49_{\pm23.66}$ | $\mathbf{21.17}_{\pm2.06}$ |
| EWP | $677.71_{\pm223.96}$ | $10.46_{\pm4.99}$ | $378.14_{\pm49.91}$ | $1.18_{\pm0.01}$ | $14.61_{\pm9.25}$ | $10.80_{\pm1.16}$ |
| MarginalIQN (N) | $1579.72_{\pm363.03}$ | $36.06_{\pm15.76}$ | $471.66_{\pm194.21}$ | $7.33_{\pm2.68}$ | $30.16_{\pm3.69}$ | $12.83_{\pm0.36}$ |
| MarginalIQN (RS) | $1169.10_{\pm173.96}$ | $13.19_{\pm4.83}$ | $207.76_{\pm92.72}$ | $3.97_{\pm3.42}$ | $29.94_{\pm4.12}$ | $12.98_{\pm0.49}$ |
| **KR-IQN(N)** | $1785.23_{\pm141.31}$ | $45.57_{\pm9.88}$ | $1070.63_{\pm152.21}$ | $15.61_{\pm0.34}$ | $\mathbf{54.97}_{\pm6.41}$ | $12.56_{\pm0.34}$ |
| **KR-IQN(RS)** | $\mathbf{1847.12}_{\pm10.60}$ | $\mathbf{49.39}_{\pm5.01}$ | $\mathbf{1163.43}_{\pm75.19}$ | $\mathbf{16.17}_{\pm0.52}$ | $49.13_{\pm4.37}$ | $12.62_{\pm0.68}$ |

*Table F.* Reference points used for hypervolume computation.

| | Cheetah | Walker2d | Hopper-2d | Ant | Hopper | Finance |
|---|---|---|---|---|---|---|
| Reference Point | $(0, -3000)$ | $(0, -500)$ | $(0, -100)$ | $(0, 0, -500)$ | $(0, -100, -100)$ | $(-100, -210, -575)$ |

2. $\text{asset}_2$ is the bond: TLT (Long-term Treasury bond ETF), TIP (Inflation-protected bond ETF), JNK (High-yield bond ETF),

3. $\text{asset}_3$ is the stock equity: SPY (S&P 500 ETF), QQQ (Nasdaq-100 ETF), SOXX (Semiconductor ETF),

4. $\text{asset}_4$ is the cash.

The coefficients 0.1 and 0.5 are hand-crafted to balance the scale across different reward components.

**Reward 3 (Logarithmic Earnings):** The third reward is defined as

$$R_3(s, a) = 100 \cdot \log \left( \frac{\text{Asset}_{t+1}}{\text{Asset}_t} \right). \tag{D.31}$$

The coefficient 100 is set to a large value since the daily log return is typically close to zero.

### D.4. Detailed Experiment Results

Tables I and J present experimental results not included in the main text. For ease of comparison, some entries overlap with Tables E and 1. Due to the negative correlation among objectives, $\text{EU}_{\text{Risk}}^{50\%}$ exceeds EU in most cases. This occurs because pessimistic outcomes on a risk-sensitive objective tend to yield optimistic outcomes on other axes; when a policy is sufficiently mature, trade-offs among objectives naturally induce negative correlations. This observation carries two implications: it suggests the insufficiency of projective TQC, and supports the importance of the simplex risk measure.

### D.5. Additional Ablation Studies

When the number of critics is 4, performance appears to decrease slightly compared to when it is 3. This seems to be due to the TQC-related ratio, although the decrease is not statistically significant. Similarly, when the number of quantiles is 18 compared to 16, the other metrics remained largely unchanged, with only $\text{EU}_{\text{Risk}}$ showing a slight increase.

*Table G.* Observation Space of the Financial Environment. (T) denotes technical indicators

| Type | Features | | | | | | | |
|------|----------|---|---|---|---|---|---|---|
| Per Asset | Open Price | Buy Avg / Open | num.Share | Prev. OHLVC | EMA (T) | RSI (T) | CCI (T) | MACD (T) |
| Common | | | | Cash | | | | |

*Table H.* Ablation results with different num. critics and num. quantiles (critic training) values.

| Num. critics | EU | $EU_{Risk}$ | $HV_{(\times 10^8)}$ | $HV_{Risk(\times 10^8)}$ |
|---|---|---|---|---|
| 2 | $1036.025_{\pm 85.448}$ | $692.215_{\pm 268.684}$ | $10.10_{\pm 1.57}$ | $9.88_{\pm 1.52}$ |
| 3 (Default) | $1253.93_{\pm 11.36}$ | $1086.62_{\pm 106.02}$ | $16.28_{\pm 0.47}$ | $15.96_{\pm 0.60}$ |
| 4 | $1184.276_{\pm 81.181}$ | $1004.179_{\pm 149.341}$ | $13.40_{\pm 2.31}$ | $13.20_{\pm 2.33}$ |

| Num. quantiles | EU | $EU_{Risk}$ | $HV_{(\times 10^8)}$ | $HV_{Risk(\times 10^8)}$ |
|---|---|---|---|---|
| 12 | $1233.629_{\pm 22.589}$ | $923.381_{\pm 313.557}$ | $15.30_{\pm 1.39}$ | $15.10_{\pm 1.41}$ |
| 14 | $1201.921_{\pm 48.807}$ | $868.066_{\pm 273.692}$ | $14.50_{\pm 1.66}$ | $14.00_{\pm 2.06}$ |
| 16 (Default) | $1253.93_{\pm 11.36}$ | $1086.62_{\pm 106.02}$ | $16.28_{\pm 0.47}$ | $15.96_{\pm 0.60}$ |
| 18 | $1245.09_{\pm 9.499}$ | $1142.541_{\pm 56.043}$ | $15.60_{\pm 0.82}$ | $15.40_{\pm 0.77}$ |

Figure D depicts the results when we shuffle the environment's objective order. As shown, while the EU shows almost no change, $EU_{Risk}$ responds somewhat more sensitively because it accounts for tail values.

### D.6. Case Study Detail

**Metrics:** Since earning is a straightforward metric, we begin by explaining the concept of MDD. Let $\{R_t\}_{t=0}^T$ be a sequence of asset price from initial time $0$ to the final time $T$. The MDD is the largest decline from the running maximum and defined as

$$\text{MDD}(\{R_t\}_{t=0}^T) = \max_{t=0,\ldots,T} \max_{s=0,\ldots,t} \left(1 - \frac{R_t}{R_s}\right). \tag{D.32}$$

The Sharpe ratio evaluates the risk-adjusted return of a strategy by comparing its excess return over the risk-free rate to its return volatility. Let $r_t$ denote the daily return at time $t = 1,\ldots,T$; i.e., $(R_t - R_{t-1})/R_{t-1}$. Let $r_f$ be risk-free interest rate (often, the base rate). Then the Sharpe ratio is

$$\text{Sharpe}(\{R_t\}_{t=0}^T) = \frac{\mathbb{E}[r_t - r_f]}{\sqrt{\mathbb{V}\text{ar}[r_t - r_f]}}. \tag{D.33}$$

Similar to Sharpe ratio, Sortino ratio also evaluates the risk-adjusted return of a strategy. However, it only accounts for lower-semi volatility;

$$\text{Sortino}(\{R_t\}_{t=0}^T) = \frac{\mathbb{E}[r_t - r_f]}{\sqrt{\mathbb{V}\text{ar}[\min(r_t, 0)]}}. \tag{D.34}$$

*Table I.* Finance Test ($i = 3$)

| Category | Name | EU | $EU_{Risk}^{50\%}$ | $EU_{Risk}^{10\%}$ | HV $(\times 10^6)$ | $HV_{Risk}^{50\%}$ $(\times 10^6)$ | $HV_{Risk}^{10\%}$ $(\times 10^6)$ |
|---|---|---|---|---|---|---|---|
| KR-IQN (ours) | CVaR | $10.72_{\pm 6.81}$ | $6.85_{\pm 6.28}$ | $6.10_{\pm 5.73}$ | $12.57_{\pm 0.26}$ | $12.56_{\pm 0.25}$ | $12.55_{\pm 0.24}$ |
| | Neutral | $10.06_{\pm 3.71}$ | $5.56_{\pm 2.69}$ | $4.05_{\pm 3.07}$ | $12.45_{\pm 0.23}$ | $12.44_{\pm 0.23}$ | $12.42_{\pm 0.23}$ |
| | $\text{Simplex}_{0.5}$ | $12.67_{\pm 4.88}$ | $7.69_{\pm 4.90}$ | $6.79_{\pm 5.15}$ | $12.76_{\pm 0.25}$ | $12.74_{\pm 0.25}$ | $12.73_{\pm 0.25}$ |
| | $\text{Wang}_{-0.71}$ | $11.75_{\pm 5.17}$ | $8.39_{\pm 4.54}$ | $7.58_{\pm 4.45}$ | $12.65_{\pm 0.23}$ | $12.63_{\pm 0.23}$ | $12.61_{\pm 0.22}$ |
| MarginalIQN | CVaR | $7.98_{\pm 4.48}$ | $6.10_{\pm 4.24}$ | $5.58_{\pm 4.38}$ | $12.30_{\pm 0.28}$ | $12.28_{\pm 0.28}$ | $12.27_{\pm 0.28}$ |
| | Neutral | $6.10_{\pm 1.09}$ | $4.14_{\pm 1.34}$ | $3.75_{\pm 1.31}$ | $12.35_{\pm 0.11}$ | $12.34_{\pm 0.11}$ | $12.33_{\pm 0.12}$ |
| | $\text{Simplex}_{0.5}$ | $6.57_{\pm 5.60}$ | $4.53_{\pm 4.94}$ | $3.93_{\pm 4.42}$ | $12.14_{\pm 0.40}$ | $12.13_{\pm 0.39}$ | $12.12_{\pm 0.38}$ |
| | $\text{Wang}_{-0.71}$ | $8.67_{\pm 4.58}$ | $4.00_{\pm 4.82}$ | $3.10_{\pm 4.30}$ | $12.40_{\pm 0.33}$ | $12.38_{\pm 0.33}$ | $12.37_{\pm 0.32}$ |
| Baselines | EWP | $2.63_{\pm 2.05}$ | $0.76_{\pm 1.03}$ | $0.60_{\pm 0.92}$ | $11.99_{\pm 0.29}$ | $11.99_{\pm 0.29}$ | $11.98_{\pm 0.29}$ |
| | PGMORL | $-0.59_{\pm 0.23}$ | $-3.09_{\pm 0.36}$ | $-6.09_{\pm 0.48}$ | $11.54_{\pm 0.04}$ | $11.36_{\pm 0.04}$ | $11.15_{\pm 0.05}$ |
| | CAPQL | $4.01_{\pm 3.73}$ | $2.75_{\pm 2.78}$ | $1.79_{\pm 1.86}$ | $11.83_{\pm 0.26}$ | $11.83_{\pm 0.26}$ | $11.82_{\pm 0.25}$ |
| | DPMORL | $3.28_{\pm 2.34}$ | $2.63_{\pm 1.98}$ | $-2.89_{\pm 1.78}$ | $13.11_{\pm 0.43}$ | $14.21_{\pm 0.83}$ | $13.35_{\pm 0.50}$ |

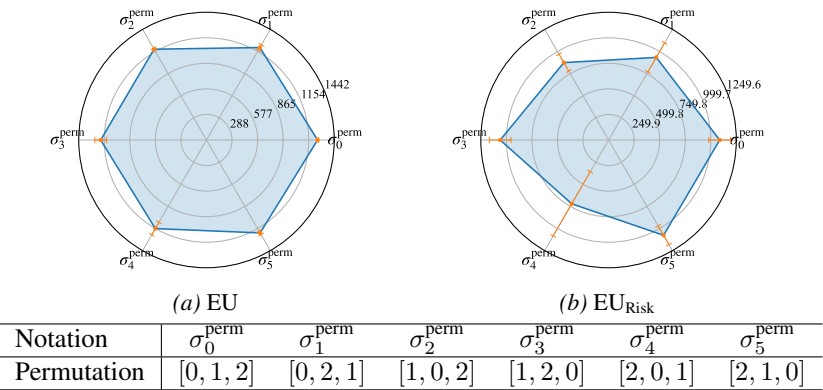

| Notation | $\sigma_0^{\text{perm}}$ | $\sigma_1^{\text{perm}}$ | $\sigma_2^{\text{perm}}$ | $\sigma_3^{\text{perm}}$ | $\sigma_4^{\text{perm}}$ | $\sigma_5^{\text{perm}}$ |
|---|---|---|---|---|---|---|
| Permutation | $[0, 1, 2]$ | $[0, 2, 1]$ | $[1, 0, 2]$ | $[1, 2, 0]$ | $[2, 0, 1]$ | $[2, 1, 0]$ |

*Figure D.* Hopper: Objective shuffling results

*Table J.* Finance Train ($i = 3$)

| Category | Name | EU | $\text{EU}_{\text{Risk}}^{50\%}$ | $\text{EU}_{\text{Risk}}^{10\%}$ | HV ($\times 10^6$) | $\text{HV}_{\text{Risk}}^{50\%}$ ($\times 10^6$) | $\text{HV}_{\text{Risk}}^{10\%}$ ($\times 10^6$) |
|---|---|---|---|---|---|---|---|
| KR-IQN (ours) | CVaR | $46.36_{\pm 3.92}$ | $49.13_{\pm 4.37}$ | $27.84_{\pm 5.86}$ | $11.87_{\pm 0.42}$ | $12.62_{\pm 0.68}$ | $12.99_{\pm 0.37}$ |
| | Neutral | $50.55_{\pm 6.87}$ | $54.97_{\pm 6.41}$ | $32.05_{\pm 6.59}$ | $11.49_{\pm 0.46}$ | $12.56_{\pm 0.34}$ | $13.12_{\pm 0.23}$ |
| | $\text{Simplex}_{0.5}$ | $50.63_{\pm 8.44}$ | $54.52_{\pm 8.34}$ | $28.84_{\pm 5.75}$ | $11.50_{\pm 0.68}$ | $12.09_{\pm 0.51}$ | $12.40_{\pm 0.33}$ |
| | $\text{Wang}_{-0.71}$ | $50.81_{\pm 10.68}$ | $55.48_{\pm 13.06}$ | $37.26_{\pm 8.09}$ | $11.91_{\pm 0.55}$ | $12.83_{\pm 0.20}$ | $13.24_{\pm 0.33}$ |
| MarginalIQN | CVaR | $50.41_{\pm 7.66}$ | $50.09_{\pm 6.62}$ | $29.94_{\pm 4.12}$ | $11.68_{\pm 0.55}$ | $12.48_{\pm 0.48}$ | $12.98_{\pm 0.49}$ |
| | Neutral | $51.44_{\pm 6.60}$ | $53.66_{\pm 4.57}$ | $30.16_{\pm 3.69}$ | $11.59_{\pm 0.80}$ | $12.28_{\pm 0.70}$ | $12.83_{\pm 0.36}$ |
| | $\text{Simplex}_{0.5}$ | $55.34_{\pm 4.46}$ | $57.48_{\pm 5.77}$ | $35.28_{\pm 2.94}$ | $12.12_{\pm 0.46}$ | $12.94_{\pm 0.35}$ | $13.18_{\pm 0.35}$ |
| | $\text{Wang}_{-0.71}$ | $50.18_{\pm 10.38}$ | $54.00_{\pm 11.05}$ | $31.15_{\pm 4.66}$ | $11.56_{\pm 1.17}$ | $12.65_{\pm 1.00}$ | $12.80_{\pm 0.86}$ |

The $\text{VaR}_{5\%}$ is defined as the return corresponding to the worst 5% quantile of outcomes, i.e.,

$$\text{VaR}_{5\%}(\{R_t\}_{t=0}^T)) = F_{R_T}^{-1}(0.05). \tag{D.35}$$

**Equity Curve Examples:** Figure E shows examples of equity curves generated by KR-IQN. As shown in Figure E(c), a policy not selected based on the Sortino ratio may still offer an advantage: very low volatility. Additionally, even a selected policy may underperform in terms of returns. In contrast, the risk-neutral approach often fails to generate such policies, as shown in Figure E(a). Conversely, in Figure E(b), the return-seeking policy (orange dashed line) initially showed the worst performance, but recovered quickly and ultimately achieved higher returns than the conservative policy.

# E. Discussion on Empirical Methodology

Risk-sensitive MORL inherently demands substantial computational cost for both training and evaluation. Moreover, methods such as DPMORL and PGMORL require separate training for each utility function. This raises two discussion points.

**Exhaustive Evaluation:** Due to the nature of risk-sensitive measures, evaluating a single policy requires 100 rollouts, multiplied by the number of simplex iterations and random seeds. We attempted to apply more than two risk-sensitive axes for evaluation, but this requires at least 10,000 samples to obtain a meaningful $\rho_{10\%}^{\text{CVaR}}$, resulting in 660,000 evaluations per seed. This is because the required number of samples increases quadratically with each additional axis. This computational burden underscores the need for discussion about more efficient evaluation methodologies in risk-sensitive MORL.

**Policy Choice:** Methods like DPMORL and PGMORL require real-environment evaluation before deployment, yet such assumptions are infeasible in domains like finance. We introduced IC as a workaround, but this does not fully address the practical limitations.

To the best of our knowledge, as a pioneering study in risk-sensitive MORL, we performed extensive empirical evaluations. Nevertheless, the associated computational burden extends beyond the scope of this paper. This challenge represents a

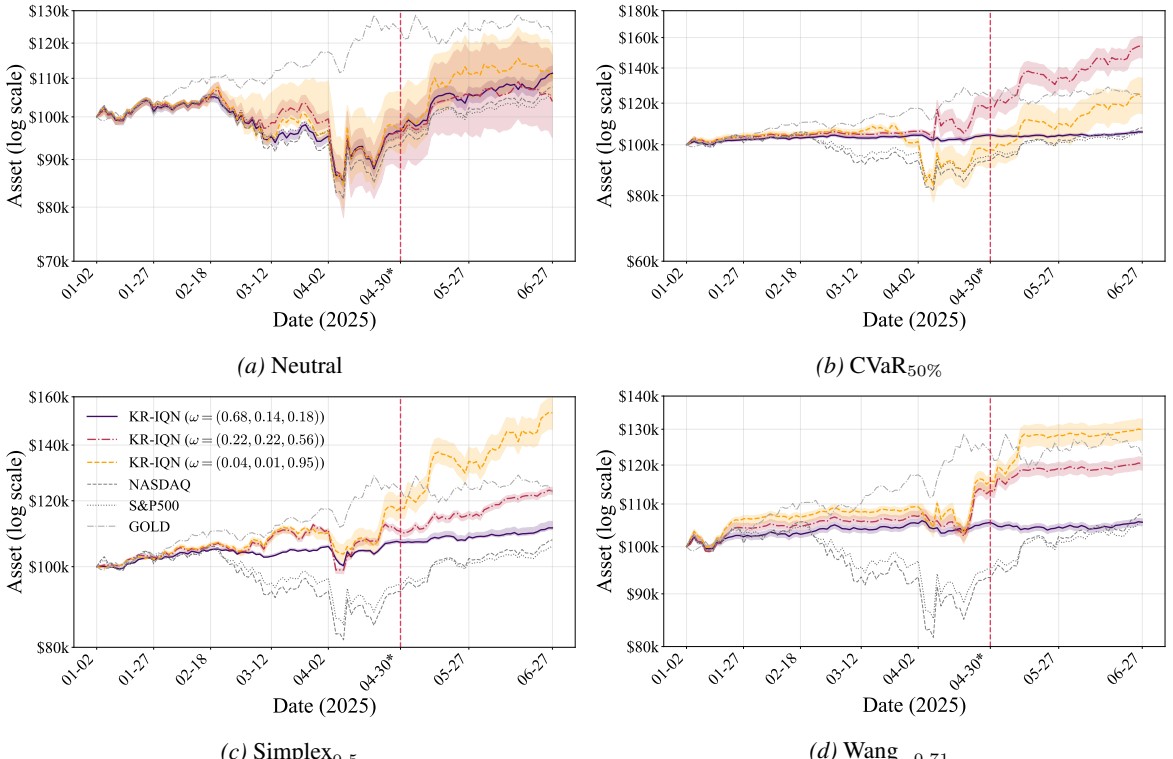

*(a)* Neutral

*(b)* CVaR$_{50\%}$

*(c)* Simplex$_{0.5}$

*(d)* Wang$_{-0.71}$

*Figure E.* Equity Curve Examples. Benchmark assets are priced via ETFs: QQQ (NASDAQ), SPY (S&P 500), and GLD (Gold). April 30th (04-30) is the policy selection date. The orange dashed lines represent return-seeking policies (weighted toward returns), the crimson dash-dotted lines represent balanced policies, and the dark violet solid lines represent conservative policies (focused on lower volatility).

broader methodological hurdle for the field rather than a deficiency in specific algorithms, including ours or baselines. Additional experiments alone cannot address these fundamental limitations; thus, the community must reach a consensus on appropriate evaluation standards given computational constraints.

We outline some possible directions to address these issues. One potential solution leverages the distributional properties of value functions.

**Conjecture E.1.** *Let $(\mathcal{S}, \mathcal{A}, \mathbf{R}, \mathcal{P}, \gamma)$ be an MO-MDP with finite state and action spaces. For $\omega \in \mathcal{W}$, suppose the optimal policy $\pi_\omega^*$ induces a recurrent Markov chain; i.e., $\Pr[\tau_{s_0} < \infty \mid s_0] = 1$ and $\mathbb{E}[\tau_{s_0}] < \infty$, where $\tau_{s_0} = \inf\{n \geq 1 : s_n = s_0\}$. Then the value distribution $\mathbf{Z}_\omega^*$ is asymptotically normal: $\mathbf{Z}_\omega^* \to \mathcal{N}(\boldsymbol{\mu}, \boldsymbol{\Sigma})$ for some $\boldsymbol{\mu} \in \mathbb{R}^d$ and covariance matrix $\boldsymbol{\Sigma} \in \mathbb{R}^{d \times d}$.*

This conjecture is a natural extension of the well-known result that if $\pi$ induces a recurrent Markov chain, then the univariate value distribution $Z^\pi$ is asymptotically normal (Mendoza-Pérez & Hernández-Lerma, 2010). We leave this as a conjecture, as a formal proof is beyond the scope of this work. Furthermore, at least for the mujoco benchmarks, the optimal policies empirically induce recurrent chains. See Figure F. Leveraging this observation, one may consider:

1. Performing statistical tests (e.g., Mardia, Energy ) to assess whether the value distribution follows a Gaussian distribution.

2. Diagonalizing the covariance matrix and computing univariate risk measures along the principal components. Some risk measures (e.g., CVaR, EVaR) have closed form formula for univariate gaussian distribution.

The first approach at least allows for rapid screening of poor policies in known environments. However, this approach requires careful recalculation to account for the effects of discounting, and cannot be applied to policies defined on MDPs for which the central limit theorem does not hold.

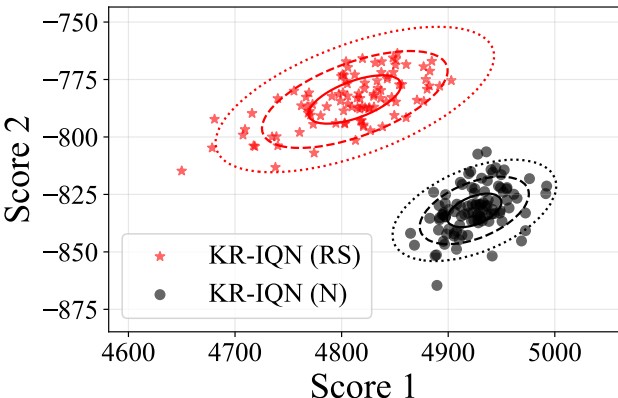

*Figure F.* Multivariate Gaussian Approximation of Cheetah Scores. The Cheetah environment is a well-known recurrent environment.

The second approach is to apply the model risk framework as discussed in (Bernard et al., 2024; Yoo & Woo, 2025). For objectives of primary interest, we can compute the empirical mean and standard deviation from rollouts; that is, we take a univariate approach along the most important objective dimension. Then, assuming the policy's value distribution follows a distribution with mean $\mu$ and standard deviation $\sigma$, we can compute the worst-case risk over all distributions whose distance from the empirical marginal distribution is at most $\varepsilon > 0$ (Bernard et al., 2024). This allows us to estimate an upper bound on risk for the objective without exhaustive evaluation.

These methods may substantially reduce the number of evaluations required, offering practical directions for future work.

