# OpenReview forum: "Knothe-Rosenblatt Quantile Regression for Risk-sensitive Multi-objective Reinforcement Learning"
_ICML.cc/2026/Conference — ICML 2026 regular_

### Official Review · Reviewer_Zppk · 2026-03-08

**Soundness:** 2
**Presentation:** 3
**Significance:** 2
**Originality:** 3
**Overall Recommendation:** 4
**Confidence:** 4

**Summary:**

This paper investigates risk-sensitive multi-objective reinforcement learning (MORL) by extending distributed reinforcement learning from scalar environments to vector-valued reward settings. The core idea is to utilize the Knothe–Rosenblatt (KR) mapping as a multivariate quantile function to model the joint reward distribution across multiple objectives. This quantile representation then enables the definition and optimization of vector risk measures in an easily interpretable manner.

More specifically, the paper formulates the objective as learning a policy for each preference vector that optimizes the vectorized risk-sensitive expectation of the multi-objective reward distribution under the KR quantile mapping. Theoretically, the authors demonstrate that the KR-based construction is consistent with the axiomatic definition of vector risk measures and further provide convergence results for KR quantile regression and adapted Wasserstein projection under distributed Bellman operators for critics.

Algorithmically, the paper introduces KR-IQN, which can be viewed as a multi-objective, risk-sensitive extension of IQN. Since the KR mapping introduces an autoregressive ordering of objectives, the method employs transformers without positional encoding and shuffles the objective order during training to mitigate induction bias caused by arbitrary ordering. The paper also introduces MO-TQC, a multi-objective variant of truncated quantile criticism that enhances training stability by truncating marginal layers and quantile atoms after scalar projection.

The method is experimentally evaluated across multiple MO-Gymnasium benchmarks and financial trading settings.

**Compliance With Llm Reviewing Policy:**

Affirmed.

**Key Questions For Authors:**

1. How sensitive is KR-IQN to the arbitrary ordering of objectives in practice, beyond the current shuffle/no-PE mitigation?
2. How much of the reported improvement comes from KR quantile modeling itself versus the additional engineering components (transformer design, shuffling, and MO-TQC)?

**Limitations:**

yes

**Strengths And Weaknesses:**

Strengths:
- The paper tackles the underexplored problem of risk-sensitive multi-objective reinforcement learning, and proposes a fairly ambitious end-to-end framework built around Knothe–Rosenblatt (KR) quantile regression.
- The central idea—using the KR map as a multivariate quantile function for MORL, then connecting it to vector-risk measures and extending IQN accordingly—is genuinely novel in this context.

Weaknesses:
- The method combines multiple linked components—KR quantile regression, transformer design choices, objective shuffling, MO-TQC, and multiple risk constructions—yet the experiments do not fully disentangle when each part is necessary or when the full framework should be preferred over simpler alternatives.
- The submission combines many ingredients—new multivariate quantile modeling, vector-risk justification, convergence analysis, a transformer estimator, MO-TQC stabilization, and several risk variants—into one paper. This makes the narrative dense and somewhat harder to evaluate cleanly.

---

> ### Author Rebuttal · Authors · 2026-03-31
>
> We appreciate your valuable feedback.
>
> **W.1.1** When the previous approach fails?
>
> As shown in Appendix B.1, the previous ***MMD-based approach has theoretical limitations*** stemming from [1], whereas the quantile regression approach in single-objective RL does not. Therefore, the problematic situation of the MMD-based approach can arise at any time, regardless of the problem setting. Furthermore, as described in Figure 1, it is ***never*** possible to conduct ***risk-sensitive RL*** with the MMD-based approach, as it lacks the transport property.
>
> Further, Marginal IQN does ***not*** guarantee the convergence of critics unless all objectives are independent; for an illustration of the dependency, please see Figure C in the Appendix.
>
> **W.1.2** Why are these components needed?
>
> **KR-Quantile Regression.**
> The natural question is: "Are we free from this theoretical problem when using the KR map?" Proposition 4.5 confirms this. The next question is: "Why the KR map over other multivariate quantile definitions?" We address this in Appendix B.2. This choice further allows us to learn a risk-sensitive actor in an IQN-like manner.
>
> **Multiple Risk Construction.** Once the KR map is chosen as a distributional critic, we must show that risk-sensitive MORL can be achieved through the KR-distributional critic, which is our final goal. This is non-trivial, as vector risk does not have an interpretable definition like the univariate case, but rather an axiomatic one. Our contribution is justifying that our approach performs genuine risk-sensitive RL, rather than serving as a proxy.
>
> **Transformer Design Choice.** To the best of our knowledge, ***the simplest approach to implementing the KR map is using a transformer***. This is because the KR map is invariant to the ordering of previously realized values. For example, with $d=3$ and realized values $z_1, z_2$: whether $(z_2, z_1)$ or $(z_1, z_2)$ is given, the map $[\mathbf{Q}_{\mathbf{Z}}(\mathbf{u}_3 \mid z_1, z_2)]_3$ must remain identical. We therefore chose a transformer with PE as the ablation model, considering an input order dependency as a controlled variable. Figure 4 shows that adding PE degrades performance to 66.9% of EU Risk and 58.3% of HV Risk.
>
> **Objective Shuffling.** This is an essential step to mitigate the ordering bias of the KR map, not merely an implementation detail. As Reviewer L6Sy noted in W.1, the KR map induces an unnatural ordering that must be addressed. Figure 4 shows that without objective shuffling, EU Risk degrades to 34.3% and HV Risk to 27.2%.
>
> **MO-TQC.** The remaining issue is training instability from overestimation bias [2, 3]. The challenge is that there is no natural total order in $\mathbb{R}^d$ for $d > 1$; e.g., $(1, 2)$ and $(2, 1)$ are incomparable. We focus on the magnitude of elements causing overestimation bias and propose marginal TQC, which truncates the vector regardless of order, and projective TQC, which truncates with respect to the preference weight.
>
> **W.2**
> As we discussed the requirements of each component in W.1, all components are essential for risk-sensitive MORL. We will add a diagram describing the dependency of each component and a guiding paragraph for each section in the Introduction.
>
> | Ablation | EU Risk ↓ | HV Risk ↓ |
> |---|---|---|
> | w/o Shuffle | 34.3% | 27.2% |
> | w/ PE | 66.9% | 58.3% |
> | Marginal TQC only | 46.0% | 35.7% |
> | Projective TQC only | 56.9% | 49.8% |
>
> **Q.1** We address this in W.1. Shuffle and no-PE are principled choices, not ad hoc mitigations: the KR map does not depend on the ordering of previously determined values, so the architecture should not impose one. Figure 4 quantifies their impact: removing shuffle degrades EU Risk to 34.3% and HV Risk to 27.2%; adding PE degrades to 66.9% and 58.3%.
>
> Please also refer to our response to Q.1 of Reviewer L6Sy. permuting the order of objectives (in env.) slightly affects performance, but the effect is not significant. Theoretically it should not, as the KR map is a pushforward map, but in practice the gradient flow slightly affect the performance.
>
> **Q.2** Other than ablation results, please note that Marginal IQN and EWP are equipped with all the same engineering structures that they can have (e.g., MO-TQC). Especially, ***Marginal IQN*** shares almost the same structure as KR-IQN, making it an ***ablation for KR-quantile itself***. Please note that the KR map cannot be constructed without transformer, as it must be an ***autoregressive map by definition***.
>
> Also, on pages 7–8, we provided an ablation study showing the results as in the table above.
>
> ### **Refs.**
> [1] Dabney et.al "Distributional reinforcement learning with quantile regression." AAAI, volume 32, 2018b.
>
> [2] Fujimoto et al.. "Addressing function approximation error in actor-critic methods." ICML. PMLR, 2018.
>
> [3] Kuznetsov et al. "Controlling overestimation bias with truncated mixture of continuous distributional quantile critics." IMCL. PMLR, 2020.

---

> > ### Author Rebuttal · Reviewer_Zppk · 2026-04-01
> >
> > I am very grateful to the authors for answering my questions. I have raised my rating accordingly.

---

> > > ### Author Response · Authors · 2026-04-02
> > >
> > > We appreciate your re-evaluation. If you have any further questions, please feel free to ask.

---

### Official Review · Reviewer_L6Sy · 2026-03-10

**Soundness:** 2
**Presentation:** 3
**Significance:** 3
**Originality:** 3
**Overall Recommendation:** 4
**Confidence:** 3

**Summary:**

This paper proposes KR-IQN, a method that learns multi-objective return distributions via Knothe–Rosenblatt (KR) quantile regression to support quantile-integral spectral risk evaluation. The paper also presents vector-risk and critic-convergence results, along with a practical transformer-based implementation and MO-TQC for improved stability and reduced overestimation in multi-objective RL (MORL).

**Compliance With Llm Reviewing Policy:**

Affirmed.

**Final Justification:**

I was initially not fully convinced that the authors’ proposed methodology adequately resolves the objective ordering issue in MORL. However, the additional experiments addressed this concern well in the rebuttal. Therefore, I will maintain my original positive score as my final recommendation.

**Key Questions For Authors:**

1. The ablation study suggests that objective shuffling and removing positional encoding is practically important, but it remains unclear whether KR-IQN is actually robust to permutations of reward dimensions. Since the benchmark tasks appear to have only 2 or 3 reward dimensions. Could it be possible to explicitly evaluate all reward-coordinate permutations and visualize the variation in performance?

2. The current experimental tasks do not seem to exhibit a clear canonical hierarchy over reward dimensions, which makes the paper’s focus understandable. At the same time, it could also be interesting to evaluate the method on multi-objective settings where such ordering is genuinely meaningful, in order to examine whether explicitly incorporating it leads to improved performance.

3. It would also be helpful to clarify what the boldface entries in Table 1 indicate.

4. I would also appreciate a brief comment on why KR-IQN appears to show relatively high variance in Earnings in Table 2.

5. Could you also provide some additional discussion of the performance differences among the Neutral, CVaR, Wang, and Simplex risk measures in Table 2?

6. The statement in Section 3.2 that the reward “may be random” seems to introduce some ambiguity regarding the independence condition used in Lemmas A.3 and A.4. It would be helpful to clarify this point.

**Limitations:**

See above.

**Strengths And Weaknesses:**

Strengths

1. The motivation of the paper is clear. Unlike prior MORL approaches based on Pareto optimality or joint distribution matching, the paper takes a direct approach to spectral-risk evaluation by learning a multivariate quantile representation from which the spectral risk functional can be computed.

2. The exposition is clear, and I found the development of the proper vector-risk measure and the convergence results appealing. The subsequent practical implementation is also reasonable.

3. The method demonstrates strong empirical performance on multi-objective RL benchmarks, and I also appreciated the additional experiments on real-world financial data using Neutral, Simplex, and Wang risk measures.

Weaknesses

1. My major concern is the use of Knothe–Rosenblatt quantile regression for the general MORL. Since KR quantile regression is intrinsically triangular and order-dependent, it is unclear whether imposing an autoregressive ordering over reward dimensions is appropriate in settings where no natural objective hierarchy exists. Although the paper mitigates this issue using objective shuffling and a transformer without positional encoding, the ordering bias does not seem to be fundamentally resolved in general.

2. While the ablation study highlights the importance of objective shuffling, it remains unclear whether the learned critic is actually robust to permutations of reward dimensions (see Key Questions 1).

---

> ### Author Rebuttal · Authors · 2026-03-31
>
> We appreciate your thoughtful feedback.
>
> **W.1** We have acknowledged that order bias is a limitation of our work, as discussed in the Conclusion and Limitations section. A promising approach to eliminate the order bias problem is optimal transport as a vector quantile, as discussed in the same section. However, OT is still not free from Bellemare's proposition (for details, please see Appendix B.1). The KR-map is the best approach we are aware of, and extending our results to OT would be an interesting direction for future work.
>
> **W.2** Since the KR-map should theoretically be a pushforward, the results should be perfectly identical; however, there may be slight variations due to the effects of gradient flow. For example, the gradient for the third value is backpropagated through the second and first values, whereas the gradient for the first value is backpropagated only once. This is consistent with the experimental results presented in Q.1.
>
> ---
> **Q.1**
> We include the results for the Hopper environment with a limited number of seeds.
> The effect of reward permutation on the environment appears to be negligible, but not entirely absent.
>
>
> | permutation | [0, 1, 2] | [2, 1, 0] | [2, 0, 1] | [1, 2, 0] | [1, 0, 2] | [0, 2, 1] |
> |---|---|---|---|---|---|---|
> | EU | 1170.76 | 1032.012 | 1203.38 | 1129.57 | 1183.0673 | 1183.06 |
> | EU RISK | 985.48 | 780.73 | 993.11 | 874.13 | 895.52 | 795.52 |
>
>
> Due to the transformer decoding steps required for critic updates in KR-IQN, training a single seed takes 4–10 hours on an RTX 5090 GPU, and evaluation alone takes approximately 40 minutes per seed (as discussed in Appendix D). Since we report results across 5 seeds, exhaustively evaluating all $d!$ permutations for multiple environments is computationally infeasible within the given timeframe. However, we will include the results in the Appendix if our paper is accepted.
>
> We will update the results as the experiments with more seeds are still in progress.
>
> **Q.2**
> To verify this, we compared KR-IQN (CVaR) in a Finance 2D environment where the first reward was removed (using only $(r_2, r_3)$). In portfolio management, the allocation plan occurs first in chronological order, and the earnings are then determined based on the actual purchase price. As expected, the earning-first ordering showed slightly better performance, but the difference was not statistically significant.
>
> The slight difference could be attributed to randomness, but it is also consistent with the gradient flow effect discussed above, where the gradient for the later value is backpropagated through earlier values, potentially introducing a subtle ordering bias.
>
> |Finance 2d | EU | EU RISK | HV | HV RISK |
> |---|---|---|---|---|
> |allocation first | 82.02 $\pm$ 3.23 | 46.57 $\pm$ 4.07 | 16.37 $\pm$ 0.46 | 14.48 $\pm$ 0.37 |
> |earning first | 85.29 $\pm$ 9.91 | 47.80 $\pm$ 3.74 | 16.86 $\pm$ 0.70 | 14.53 $\pm$ 0.19 |
>
> **Q.3** The bold notation in Table 1 indicates the best performance among the baselines. We will clarify this in the table caption and ensure consistent formatting across all tables.
>
> **Q.4** As can be observed across other baselines, the absolute value of earnings and the standard deviation tend to be proportional (a scale effect). Since KR-IQN achieved higher earnings, the standard deviation is naturally larger as well. We observed the same phenomenon in the new baselines (Zhang's MMD approach) in the response to Reviewer T4aS (18.61 $\pm$ 15.91).
>
> **Q.5** As shown in Appendix Figure B, how the agent responds to returns driven by the market (a factor known as beta in finance) varies depending on the risk sensitivity. In particular, Neutral tends to pursue more beta exposure, while Simplex follows beta with a smaller allocation. CVaR and Wang were observed to respond more to surge signals rather than beta. We will add a discussion of these results in Appendix C.5.
>
> **Q.6** This follows from the standard formulation in [1] page 4. Specifically, when the reward is assumed to depend only on $(s, a)$ (as in the standard MDP formulation), conditional independence given $(s_t, a_t)$ holds. We will add a clarification of this point in Appendix A.2.
>
> [1] Bellemare, Marc G., Will Dabney, and Rémi Munos. "A distributional perspective on reinforcement learning." International conference on machine learning. Pmlr, 2017.

---

> > ### Author Rebuttal · Reviewer_L6Sy · 2026-04-03
> >
> > I appreciate the authors’ thoughtful response and I maintain my positive score.

---

> > > ### Author Response · Authors · 2026-04-06
> > >
> > > We sincerely thank the reviewer for the encouraging feedback and keen interest in our work.
> > > As suggested, a spider chart illustrating the reward order permutations will be included in the revision, along with some minor revisions in Appendix A.2 and presentations.

---

### Official Review · Reviewer_T4aS · 2026-03-16

**Soundness:** 2
**Presentation:** 2
**Significance:** 3
**Originality:** 3
**Overall Recommendation:** 3
**Confidence:** 5

**Summary:**

This paper extends distributional reinforcement learning to the setting of risk-sensitive multi-objective reinforcement learning by introducing the Knothe–Rosenblatt (KR) map as a multivariate quantile formulation for return distributions. Building on the observation that univariate quantile functions enable spectral risk measures in single-objective RL, the authors use the KR map to define a proper vector-risk measure that can capture dependencies across objectives, unlike marginal-based approaches, while avoiding the lack of transport structure associated with MMD-based methods. On the theoretical side, the paper formulates KR quantile regression under the adapted Wasserstein distance and presents a contraction result for the corresponding distributional multi-objective Bellman operator. On the algorithmic side, the authors note that the conditional nature of the KR map introduces sensitivity to objective ordering, and they mitigate this issue by using a transformer-based critic without positional encoding together with random shuffling of objective order during training, yielding the proposed KR-IQN architecture, further combined with a multi-objective variant of TQC for improved stability. Experiments on MO-Gymnasium benchmarks and a financial trading environment show that the proposed method performs competitively against several existing MORL baselines, with particularly strong results on risk-sensitive evaluation metrics.

**Compliance With Llm Reviewing Policy:**

Affirmed.

**Final Justification:**

I am maintaining my score, as the authors' clarification that "Marginal-IQN with transformer" is structurally impossible does not resolve the core concern: the paper provides no ablation comparing the transformer against simpler permutation-invariant aggregators (e.g., DeepSets or sorted-concat MLP) for implementing the KR map, leaving it unclear whether the performance gains stem from the KR formulation itself or from the expressive power of the transformer architecture specifically.

**Key Questions For Authors:**

- *Could the authors report the parameter size of the KR-IQN critic head, and ideally the total parameter counts of KR-IQN versus Marginal-IQN/EWP?* Since the paper states that these methods share the same base architecture and mainly differ in the critic head, this would help clarify whether the gains come from the KR formulation itself or partially from increased model capacity.
- *Could the authors provide learning curves over training iterations/environment steps for the main benchmarks?* The current presentation focuses mostly on final performance tables, making it difficult to assess convergence speed, optimization stability, and sample efficiency.

**Limitations:**

The main limitations are clarity, theory presentation, and experimental positioning. In particular, key definitions and figures are under-explained, the connection between the adapted Wasserstein contraction result and the final KR-IQN / MO-TQC implementation is not fully transparent, and the empirical comparison could be strengthened with more explicit coverage of recent relevant baselines.

**Strengths And Weaknesses:**

## Strengths

1. **Well-motivated problem formulation.** The paper is well motivated in introducing a KR-map-based formulation for risk-sensitive MORL. In particular, the motivation against prior MMD-based multivariate distributional methods is clear: the authors argue that MMD-based approaches do not provide an order-preserving transport map, while marginal-based methods fail to capture inter-objective dependency. Framing KR quantile regression as a way to preserve transport structure while modeling dependencies across objectives is a meaningful and interesting idea.
2. **Reasonable mitigation of KR ordering sensitivity.** A central practical issue of the KR map is its dependence on objective ordering. The proposed architectural choice—using a transformer critic without positional encoding and randomly shuffling the objective order during training—is a sensible way to reduce this artificial inductive bias. Even if not fully eliminating the issue, this is a thoughtful design choice that is well aligned with the main methodological challenge identified by the paper.



## Weaknesses

1. **The exposition is difficult to follow in several core places.** Some of the most important definitions and figures are introduced too abruptly, without enough clarification of notation or intuition. For example, the paper would benefit from a cleaner explanation of the KR map’s domain/codomain transformation, clearer notation around Eq. (4), and a more explicit walkthrough of the semantics of the inputs in Figure 3. As written, several key concepts that are central to understanding the method are left under-explained.
2. **The presentation under-emphasizes the paper’s main technical contribution.** The paper spends substantial space on general background, while some of the most important technical ingredients needed to assess the main claim are either brief or deferred to the appendix. In particular, the adapted Wasserstein distance, the AW projection, and the proof structure behind Proposition 4.5 are not explained with enough detail in the main paper, even though Proposition 4.5 appears to be one of the core theoretical results. This makes it harder than necessary to evaluate the actual novelty and scope of the theorem. The paper would benefit from a substantial rewrite of the theory presentation and a clearer separation between recalled background results and the authors’ own contributions.
3. **The theory-to-algorithm connection remains unclear.** While the contraction result is presented as support for the proposed framework, the connection between Proposition 4.5 and the final KR-IQN / MO-TQC implementation is not sufficiently transparent. In particular, the paper does not clearly explain how the practical parameterization and truncation mechanisms relate to the theoretical object being analyzed. As a result, it is difficult to tell how much of the final algorithm is covered by the stated theory versus added as an empirical engineering layer.
4. **The experimental comparison may be incomplete relative to recent literature.** The experimental section compares against PGMORL, DPMORL, CAPQL, EWP, and Marginal-IQN, which are reasonable baselines, but the paper would be stronger if it more explicitly positioned itself against the most recent MORL and risk-sensitive/distributional RL literature. As written, it is not fully clear whether the empirical comparison reflects the strongest currently relevant alternatives or mainly a subset of standard baselines.

- Wiltzer, H., Farebrother, J., Gretton, A., & Rowland, M. (2024). *Foundations of multivariate distributional reinforcement learning*. *Advances in Neural Information Processing Systems, 37*, 101297–101336.
- Yoo, G., Park, J., & Woo, H. (2024). *Risk-conditioned reinforcement learning: A generalized approach for adapting to varying risk measures*. *Proceedings of the AAAI Conference on Artificial Intelligence, 38*, 16513–16521.
- Yoo, G., & Woo, H. (2025). *Model risk-sensitive offline reinforcement learning*. *The Thirteenth International Conference on Learning Representations*.
- Zhang, P., Chen, X., Zhao, L., Xiong, W., Qin, T., & Liu, T.-Y. (2021). *Distributional reinforcement learning for multidimensional reward functions*. *Advances in Neural Information Processing Systems, 34*, 1519–1529.
- Cai, X., Zhang, P., Zhao, L., Bian, J., Sugiyama, M., & Llorens, A. (2023). *Distributional pareto-optimal multi-objective reinforcement learning*. *Proceedings of the 37th Annual Conference on Neural Information Processing Systems*.

---

> ### Author Rebuttal · Authors · 2026-03-31
>
> We appreciate your meaningful feedback.
>
> **W.1** The KR map without any dependencies is defined as $\mathbf{Q}_{\mathbf{Z}}: [0,1]^{d} \rightarrow \mathbb{R}^{d}$ (Eq. 4).
>
> Regarding Figure 3, F.F. Emb refers to Fourier Feature Embedding, which replaces Dabney et al.'s cosine embedding [1] (see [2] for details). Lin. denotes a simple linear layer, and L.N. stands for Layer Normalization. We will add descriptions of these abbreviations to the Figure 3 caption and correct the typos (e.g., $p_{ij} \mapsto u_{ij}$).
>
> **W.2**
> We will clarify the roles of the Adapted Wasserstein distance and $\Pi_{AW}$ below Proposition 4.5.
>
> The contribution of Proposition 4.5 is proving the validity of KR-quantile regression loss for training the critic, not proposing novel proof mechanics. While the proof builds upon [3], transitioning to the KR-map setting is non-trivial, as the standard Wasserstein distance and quantile projection must be replaced with their adapted counterparts ($\Pi_{AW}$), since vector quantiles do not coincide with the Wasserstein distance when $d > 1$.
>
> **W.3**  As noted in W.2, Proposition 4.5 concerns only the critic loss, not the TQC or transformer mechanism.
>
> The transformer is used because the order of previously sampled values entering the KR map is irrelevant; an RNN or positional encoding would impose a stronger ordering bias than the KR map assumes. Meanwhile, the TQC was introduced to mitigate overestimation bias [4], which naturally exists in RL; though specialized methods are required for distributional MORL (MOTQC), they are not relevant to Proposition 4.5.
>
> **W.4**
> Following the reviewer's feedback, we report Zhang et al.'s MMD-based results (HV scales follow Table 1). It shows weaker performance on mo-gym but is competitive with KR-IQN (Neutral) on the finance test, while underperforming the risk-sensitive KR-IQN.  We will also add an extended related work in the Appendix per the reviewer's suggestion, clarifying our positioning in the literature.
>
> | Env | eu | eu_risk | hv | hv_risk |
> |---|---|---|---|---|
> | hopper | 436.36  | 172.01  | 1.73 | 1.51  |
> | hopper2d | 668.08  | 169.63  | 0.53 | 0.41  |
> | halfcheetah | 2316.82  | 2398.52 | 12.74  | 13.07 |
> | walker2d | 446.83  | 216.10 | 4.44  | 4.41  |
> | ant | 733.07  | 482.20  | 8.03| 5.96  |
> | finance | 47.81  | 26.43  | 12.82  | 13.10 |
>
> | Earning | Sharpe | Sortino | VaR | CVaR | MDD | IC |
> |---|---|---|---|---|---|---|
> | 18.61 | 1.98 | 4.46 | 17.33 | 17.27 | -0.15 | -1.25 |
>
> Please note that our baselines were selected with considerable care, guided by where our method sits within the literature. Zhang et al. introduced the initial concept, later extended by EWP (Willtzler et al.); both were included to illustrate the limitations of MMD in distributional MORL [Related Work: lines 90–95]. DPMORL (Cai et al.) optimizes over diverse utility functions to achieve first-order stochastic dominance without a distributional critic. CAPQL was adopted as the base algorithm underlying KR-IQN, Marginal IQN, and EWP, while PGMORL was included as the archetypal method for constructing a DPMOR [lines 95–100]. Furthermore, Yoo et al. address single-objective risk-sensitive RL and cannot be directly transferred to the MORL setting.
>
> **Q.1** Actually, EWP ($1.1 \times 10^6$) has 10% more parameters than KR-IQN ($1.0 \times 10^6$), and Marginal IQN ($8.9 \times 10^5$) has 11% fewer, originating from the transformer and value encoding network used only in KR-IQN (Val Enc and Transformer in Fig. 3); the parameter difference is negligible.
>  This coincides with [4], where naively increasing parameters does not yield direct improvement in RL.
>
> **Q.2** We will add this in Appendix A.2.
> We present tabular summaries of the learning curves for two environments due to character limits. Scores denote the sum of episodic rewards averaged over a sliding window of 100 episodes. Note that because the preference weight is randomly sampled each training time; i.e., scores may appear to plateau while the policy is still improving.
>
> | Hopper | T_1 | T_2 | T_3 | T_4 | T_5  ($\approx 10^6$ steps)  |
> |---|---|---|---|---|---|
> | Score 1 | 18.02 | 548.36 | 1032.36 | 1285.40 | 1290.30 |
> | Score 2 | 15.12 | 455.11 | 870.54 | 1306.84 |  1373.30 |
> | Score 3 | -0.78 | 82.30 | 132.11 | 134.30 | 141.11 |
>
> | Walker2d | T_1 | T_2 | T_3 | T_4 | T_5 ($\approx 10^6$ steps) |
> |---|---|---|---|---|---|
> | Score 1 | 4.63 | 611.86 | 1052.64 | 1035.70 | 1224.46 |
> | Score 2 | -26.28 | 208.65 | 110.23 | 48.65 | 96.93 |
>
> [1]  Dabney  et al. Implicit quantile networks for distributional reinforcement learning. ICML
>
> [2] Yoo & Woo  Model risk-sensitive offline reinforcement learning. ICLR
>
> [3] Dabney et al. Distributional reinforcement learning with quantile regression.  AAAI
>
> [4] Fujimoto et al.   "Addressing function approximation error in actor-critic methods." ICML 2018
>
> [5] Sinha  et al. "D2rl: Deep dense architectures in reinforcement learning." arXiv 2020

---

> > ### Author Rebuttal · Reviewer_T4aS · 2026-04-02
> >
> > Thank you for the response. I am maintaining my score.
> >
> > The critical issue remains unaddressed: the authors confirm that the transformer is used exclusively in KR-IQN, while all baselines use standard MLPs. This means the performance gap may stem from the transformer architecture rather than KR quantile regression itself. The key missing ablation — Marginal-IQN with the same transformer (no PE) — is absent, making it impossible to isolate the contribution of the KR formulation. The response on parameter counts does not resolve this. Other concerns (W.1, W.2, Q.2) were acknowledged but not substantively addressed in the response.

---

> > > ### Author Response · Authors · 2026-04-04
> > >
> > > We appreciate the feedback and apologize for the insufficient explanation due to character limits.
> > >
> > > ---
> > > ### **W.1**
> > >
> > > We agree that the KR map is difficult to understand.
> > >
> > > Informally speaking, the ***KR map is an autoregressive quantile representation*** of a random vector [as stated in line 80 of the introduction] with respect to the value representation (coordinate process), not previous MDP time steps. The details will be explained in the revision.
> > >
> > > Formally, let $\mathbf{X} = [X\_1, X\_2, \dots, X\_d]$ be a random vector. Consider a map $F^{-1}\_{X\_1}: [0,1] \rightarrow \mathbb{R}$, which is the marginal quantile function of $X\_1$, the first component of $\mathbf{X}$.
> > >
> > > Next, consider a map $F^{-1}\_{X\_k}: [0,1] \times \text{spt}(X\_{<k}) \rightarrow \mathbb{R}$, which represents the conditional quantile function $F^{-1}\_{X\_k \mid X\_{<k}}: [0,1] \rightarrow \mathbb{R}$; i.e., $\mathbf{X}\_{<k}:= [X\_1, \dots,  X\_{k-1}]$ as an input to represent a conditional quantile.
> > > As a previous realization of $\mathbf{X}\_{<k}$ is given as an input, the function trained with quantile regression loss will converge to the conditional quantile.
> > > Here $\text{spt}(\cdot)$ is the support of $\mathbf{X}\_{<k}$.
> > >
> > > \noindent Since the output of the preceding components serves as the input to the $k$-th quantile function, this naturally constitutes an autoregressive map with respect to the previous value.  We will emphasize this in the definition.
> > >
> > > ---
> > > ### **Marginal-IQN with the same transformer (no PE)**
> > >
> > > We respectfully clarify that it is ***impossible to implement Marginal IQN with a transformer***, as the marginal IQN does not have autoregressive properties. ***If so***, it would simply ***become KR-IQN***; as it would receive a previous value as an input and be learned through the quantile-regression loss for each component (objectives).
> > >
> > > **A. What is the Marginal IQN?** In detail, Marginal IQN consists of independent marginal quantiles for each component;
> > > i.e., $\mathbf{Q}^{\text{marg.}}\_{\mathbf{X}}: [0,1]^d \rightarrow \mathbb{R}^d$,
> > > where each component is defined as $[\mathbf{Q}^{\text{marg.}}\_{\mathbf{X}}(\mathbf{u})]\_{k} = F^{-1}\_{X_k}(u\_k)$.
> > >
> > > **B. What is the transformer responsible for?**
> > >
> > > As described in Figure 3, the transformer is only applied to the value embedding because it should be able to encode the value information in an autoregressive way to represent the conditional quantile of the $k$-th objective given that the $1 \dots, (k-1)$-th objectives' state-action values are provided, and Marginal IQN must not have this information by DEFINITION.
> > >
> > > Further, we would like to clarify a possible misunderstanding: ***Even KR-IQN does not use a transformer for the state-action embedding*** (please refer Figure 3).
> > >
> > >  **C. Why Transformer?** As pointed out in the rebuttal to Reviewer Zppk, ***the transformer is the simplest way to implement a KR map***.
> > > This is because the KR map is invariant to the ordering of previously realized values. For example, with $d = 3$ and realized values $z_1, z_2$: whether $(z_2, z_1)$ or $(z_1, z_2)$ is given, the map $[\mathbf{Q}_z(\mathbf{u}_3 \mid z_1, z_2)]_3$ must remain identical.
> > >
> > > In summary,
> > >
> > > 1. The KR-IQN critic infers state-action values for each objective sequentially (within a single MDP-step). The transformer encodes the autoregressive dependency on previously realized values.
> > >
> > > 2. The Marginal IQN (or MMD-based approaches) critic infers values for all objectives at once, therefore,  the **VALUE EMBEDDING MUST BE ABSENT** (including the transformer layer),  making it impossible to apply a transformer in a comparable manner.
> > >
> > > ---
> > > ### **W.2**
> > >
> > > We agree with that, but please be aware that providing full proof in the main text is not possible because of the page limit. We will provide the summarized contents of Appendix A.1 below Proposition 4.5, which introduces AW distance and AW-projection. Further, we will provide a sketch of the proof that
> > >
> > > 1. AW distance has the partition property (Lemma A.3)
> > >
> > > 2. $\Rightarrow \gamma$-contraction property of the Bellman equation **without** AW projection,
> > >
> > > 3. $\Rightarrow \gamma$-contraction property of the Bellman equation **with** AW projection.
> > >
> > >
> > >
> > > ### **Q.2.**
> > >
> > > We provide the learning curve:
> > > https://anonymous.4open.science/r/icmlrebuttal-5F22/LearningCurves.pdf
> > >
> > > ---
> > > I hope this rebuttal resolves all concerns of the reviewer, and please reconsider the evaluation.
> > >
> > > Best regards,
> > >
> > > The authors.

---

### Official Review · Reviewer_pvDU · 2026-03-16

**Soundness:** 2
**Presentation:** 2
**Significance:** 4
**Originality:** 3
**Overall Recommendation:** 5
**Confidence:** 3

**Summary:**

This paper proposes a framework for risk-sensitive multi-objective RL that extends IQN-style distributional RL to vector-valued returns using the Knothe-Rosenblatt (KR) map as a multivariate quantile function. The authors show that the KR map induces a valid vector-risk measure satisfying zero, monotonicity, and translation equivariance (Theorem 4.2), and that a distributional Bellman operator with an adapted Wasserstein projection is a gamma-contraction (Proposition 4.5). They also propose a "simplex risk measure" that goes beyond standard CVaR. On the implementation side, they use a transformer without positional encoding to handle the artificial objective ordering that comes with the KR map's autoregressive structure, and introduce MO-TQC with marginal and projective truncation for overestimation bias. Experiments across several environments show improvements over baselines on risk-sensitive metrics.

**Compliance With Llm Reviewing Policy:**

Affirmed.

**Final Justification:**

The rebuttal addressed my two main empirical concerns (ablation scope and scaling beyond d=3) with new results, and was transparent about the partial significance-testing picture. I am updating my overall recommendation from 4 to 5. Confidence remains at 3.

I recommend acceptance. The paper makes a solid, theoretically grounded contribution to risk-sensitive multi-objective RL that the community will build on, and the authors have demonstrated responsiveness to reviewer feedback.

**Key Questions For Authors:**

1. **Ablation Studies. ** See Weakness 1. Have you run ablations on d=3 environments? How does performance change as d goes beyond 3? I will willing to increase my score if more comprehensive ablation studies are provided.

2. **Statistical isgnificance.** Have you run significance tests on the main results? What fraction of reported improvements are significant at p < 0.05? If the risk-sensitive metrics show consistently significant gains while the standard metrics don't, that would actually help clarify what this method is really buying you.

**Limitations:**

Yes.

**Strengths And Weaknesses:**

Strengths:

1, The theoretical connection between the KR map and vector-risk measures is solid. Theorem 4.2 shows the KR-based risk measure satisfies all three vector-risk axioms from Definition 4.1, and Proposition 4.5 gives convergence of the distributional Bellman operator under the adapted Wasserstein distance. This is harder than it might look. I also liked the observation that MMD-based methods lack the transport property needed for risk calculation, and the connection to [1] showing that standard convex vector-risk measures are "separable" and miss dependence structures. This gives a clear motivation for the KR map's autoregressive conditioning.

2, The experiments cover a decent range: six MO-Gymnasium environments plus a Finance environment with real market data.

Weakness:

1, My main concern is that the ablation study only covers Hopper with 2 objectives. That's not enough to conclude that the architectural choices generalize. There are also several hyperparameters that go unexplored: number of quantile samples, number of critic ensembles (C=3).


references:

[1] On the Separability of Vector-Valued Risk Measures

[2] Foundations of Multivariate Distributional Reinforcement Learning

---

> ### Author Rebuttal · Authors · 2026-03-31
>
> We appreciate your constructive feedback.
>
> **W.1.1 Ablation study only covers Hopper with 2 objective**:
>
> Hopper is a $d=3$ environment and is distinct from Hopper2d, as shown in the Main Table. We will clarify this in the Figure 4 caption by explicitly stating Hopper $(d = 3)$.
> For the more comprehensive survey, Reviewer L6Sy requested the permutation of the model according to the permutation of the reward structure. Please refer the rebuttal for Q.1 of Reviewer L6Sy if you want to see the details.
>
> **W.2. Some hyperparameters are unexplored**
>
> Due to limited time and resources, we present results for a small number of seeds and samples here. We will update with full results as they become available, and we promise to include the complete results in the Appendix of the camera-ready version if accepted.
>
> | n_taus | 14 | 16 | 18 |
> | --- | --- | --- | --- |
> | EU | 1109.74 | 1107.29 | 1149.00 |
> | EU Risk | 914.62 | 960.98 | 1005.31 |
> | HV (10^8) | 12.43 | 15.88 | 15.51 |
> | HV Risk (10^8) | 12.38 | 15.72 | 15.44 |
>
> | n_critic | 2 | 3 | 4 |
> | --- | --- | --- | --- |
> | EU | 1080.97 | 1107.29 | 1217.46 |
> | EU Risk | 690.55 | 960.98 | 1092.77 |
> | HV  (10^8) | 10.86 | 15.88 | 15.11 |
> | HV Risk (10^8) | 10.66 | 15.72 | 15.08 |
>
> We suspect that the decrease in HV (and resp. HV risk) of critic = 4 or taus = 18  is not a performance degradation but rather noise due to insufficient seeds, which we expect to be resolved with more training runs.
>
> **Q.1. How does performance change as d goes beyond 3?**
>
> To best of our knowledge, there are no well-known benchmarks with continuous actions for environments where $d > 3$. Nevertheless, we conducted experiments on Fruit Tree ($d = 6$) using a DQN variant and observed a higher Hypervolume of 9609.62 ± 548.61 compared to [1] (9302.28) and [2] (9299.15).
>
> **Q.2 Statistical significance**
> Currently, about 30% of the reported improvements in risk-sensitive metrics show statistical significance at p < 0.05 under Welch's t-test in MuJoCo environments. This is because we used 5 seeds following previous literature (e.g., [3-6]). Excluding HalfCheetah, where scores are already saturated, we expect this fraction to increase to approximately 75% with more seeds in MuJoCo environments. This is because the ***maximum*** p-value among the remaining environments is around 0.2, and under the assumption that the underlying distribution is maintained, increasing the number of seeds would bring it below 0.05.
>
> ### **Refs.**
>
> [1] Liu, Erlong, et al. "Pareto set learning for multi-objective reinforcement learning." Proceedings of the AAAI Conference on Artificial Intelligence. Vol. 39. No. 18. 2025.
>
> [2] Basaklar, Toygun, Suat Gumussoy, and Umit Ogras. "PD-MORL: Preference-Driven Multi-Objective Reinforcement Learning Algorithm." The Eleventh International Conference on Learning Representations.
>
> [3] Dabney, Will, et al. "Implicit quantile networks for distributional reinforcement learning." International conference on machine learning. PMLR, 2018.
>
> [4] Li, Alexander, and Deepak Pathak. "Functional regularization for reinforcement learning via learned fourier features." Advances in Neural Information Processing Systems 34 (2021): 19046-19055.
>
> [5] Shen, Siqi, et al. "RiskQ: risk-sensitive multi-agent reinforcement learning value factorization." Advances in Neural Information Processing Systems 36 (2023): 34791-34825.
>
> [6] Yoo, Gwangpyo, and Honguk Woo. "Model Risk-sensitive Offline Reinforcement Learning." The Thirteenth International Conference on Learning Representations. 2025.

---

> > ### Author Rebuttal · Reviewer_pvDU · 2026-04-03
> >
> > Thank you for addressing my questions. I will slightly increase my score.

---

> > > ### Author Response · Authors · 2026-04-06
> > >
> > > We deeply appreciate Reviewer pvDU's constructive feedback and their willingness to re-evaluate the initial Weak Accept score. If permitted, the revised manuscript will include a more comprehensive investigation of the p-values (e.g., by increasing the number of random seeds) and an expanded hyperparameter search.

---

### Decision · Program_Chairs · 2026-04-30

**Decision:**

Accept (regular)

**Comment:**

This submission received generally positive support from the reviewers, who agreed that it tackles an important and underexplored problem at the intersection of multi-objective and risk-sensitive reinforcement learning. The main strengths identified were the novelty of using the Knothe–Rosenblatt map to construct a multivariate quantile representation, the theoretical grounding connecting this construction to vector-risk measures and critic convergence, and a solid empirical evaluation across MO-Gymnasium and finance settings. Several reviewers also found the rebuttal effective: it clarified the motivation for the KR formulation, added evidence on permutation sensitivity and additional ablations, and addressed concerns about statistical significance and scaling beyond the standard benchmarks.

The main remaining concerns are about clarity and isolation of contributions. In particular, one reviewer remained unconvinced that the empirical gains can be cleanly attributed to the KR formulation rather than to the transformer-based implementation, and multiple reviewers noted that the presentation of the theory and its connection to the final algorithm should be sharpened. That said, the overall balance of reviews favors acceptance: the paper appears technically meaningful, original, and likely to stimulate follow-up work, while the identified weaknesses seem more about exposition and incomplete disentanglement than about a fundamental flaw in the central idea.